# Multiomics reveals microbial metabolites as key actors in intestinal fibrosis in Crohn's disease

Xuehua Li [1,4], Shixian Hu [2,3,4], Xiaodi Shen [1,4], Ruonan Zhang [1,4], Caiguang Liu [2], Lin Xiao [2], Jinjiang Lin [1], Li Huang [1], Weitao He [1], Xinyue Wang [1], Lili Huang [1], Qingzhu Zheng [1], Luyao Wu [1], Canhui Sun [1], Zhenpeng Peng [1], Minhu Chen [2], Ziping Li [1], Rui Feng [2], Yijun Zhu [2,3], Yangdi Wang [1,5✉], Zhoulei Li [1,5✉], Ren Mao [2,5✉] & Shi-Ting Feng [1,5✉]

## Abstract

**Intestinal fibrosis is the primary cause of disability in patients with Crohn's disease (CD), yet effective therapeutic strategies are currently lacking. Here, we report a multiomics analysis of gut microbiota and fecal/blood metabolites of 278 CD patients and 28 healthy controls, identifying characteristic alterations in gut microbiota (e.g., *Lachnospiraceae*, *Ruminococcaceae*, *Muribaculaceae*, *Saccharimonadales*) and metabolites (e.g., *L*-aspartic acid, glutamine, ethylmethylacetic acid) in moderate-severe intestinal fibrosis. By integrating multiomics data with magnetic resonance enterography features, putative links between microbial metabolites and intestinal fibrosis-associated morphological alterations were established. These potential associations were mediated by specific combinations of amino acids (e.g., *L*-aspartic acid), primary bile acids, and glutamine. Finally, we provided causal evidence that *L*-aspartic acid aggravated intestinal fibrosis both in vitro and in vivo. Overall, we offer a biologically plausible explanation for the hypothesis that gut microbiota and its metabolites promote intestinal fibrosis in CD while also identifying potential targets for therapeutic trials.**

**Keywords** Crohn's Disease; Fibrosis; Gut Microbiota; Metabolites; Magnetic Resonance Enterography
**Subject Categories** Digestive System; Microbiology, Virology & Host Pathogen Interaction

## Introduction

Crohn's disease (CD) is a prevalent form of inflammatory bowel disease that affects millions of individuals worldwide (Kaplan, 2015; Torres et al, 2017). Intestinal fibrosis, a common complication of CD, causes over 50% of patients to experience refractory stricture or penetrating diseases in this organ that contribute to disability and significant healthcare expenses over the course of the illness (D'Alessio et al, 2022). However, the mechanisms of fibrogenesis remain elusive, limiting the development of effective therapeutic strategies. The gut microbiota and its functional metabolites play crucial roles in the onset of CD (Lloyd-Price et al, 2019) and may also be involved in the development of intestinal fibrosis. While the associations of the microbiota and its metabolites in fibrogenesis have been extensively studied in other organs, such as the liver (Loomba et al, 2017; Lee et al, 2020; Kwan et al, 2022), it has not yet been thoroughly explored in the gut where fibrosis is only beginning to receive attention. In a limited number of animal studies, the gut microbiota and metabolites have been reported to exacerbate or ameliorate intestinal fibrosis (Jacob et al, 2018; Ellermann et al, 2019; Xu et al, 2023). However, such data are conspicuously absent from fibrosis studies on patients with CD, and whether these micro-factors can induce macro-alterations in bowel morphology (which are often associated with clinical symptoms in patients and detectable by magnetic resonance entero-graphy [MRE]) remains unknown. To address this knowledge gap, we studied the fecal microbiome, fecal and blood metabolomes, and MRE features of 278 CD patients with variable degrees of intestinal fibrosis stratified by magnetization transfer imaging (MTI), alongside analyzing the corresponding fecal and blood data from 28 healthy controls, to elucidate the correlation between the gut microbiota and metabolites with intestinal fibrosis. Subsequently, the causal relationship between one specific metabolite (*L*-aspartic acid) and intestinal fibrosis was further validated both in vitro and in vivo, providing potential novel targets for managing intestinal fibrosis in patients with CD.

## Results

### Microbial and metabolome traits identified by multiomics vary with the severity of intestinal fibrosis

To clarify the roles of gut microbiota and metabolites in the development of intestinal fibrosis, we first investigated specific

[1]Department of Radiology, The First Affiliated Hospital, Sun Yat-Sen University, 58 Zhongshan II Road, 510080 Guangzhou, People's Republic of China. [2]Department of Gastroenterology, The First Affiliated Hospital, Sun Yat-Sen University, 58 Zhongshan II Road, 510080 Guangzhou, People's Republic of China. [3]Institute of Precision Medicine, The First Affiliated Hospital, Sun Yat-Sen University, 58 Zhongshan Road 2nd, 510080 Guangzhou, Guangdong, People's Republic of China. [4]These authors contributed equally: Xuehua Li, Shixian Hu, Xiaodi Shen, Ruonan Zhang. [5]These authors contributed equally: Yangdi Wang, Zhoulei Li, Ren Mao, Shi-Ting Feng. ✉E-mail: wangyd83@mail.sysu.edu.cn; lizhlei3@mail.sysu.edu.cn; maor5@mail.sysu.edu.cn; fengsht@mail.sysu.edu.cn

microbial and metabolomic signatures associated with this condition. We recruited a derivation cohort consisting of 214 patients with CD and 28 healthy controls (HCs) (Table EV1). The patients were stratified according to the degree of intestinal fibrosis, which was determined based on MTI findings (a validated imaging technique for accurately assessing intestinal fibrosis and unaffected by concurrent inflammation (Li et al, 2018)), and categorized as having either non-mild bowel fibrosis (BF1, $n = 93$) or moderate-severe bowel fibrosis (BF2, $n = 121$) (see "Methods"; Fig. EV1A,B). The HCs were also clinically examined to rule out any intestinal diseases.

Then, to investigate the microbial and metabolomic differences between BF1, BF2, and HCs, we performed 16S rRNA sequencing of fecal samples and paired targeted metabolomics measurements on both fecal and blood samples (Datasets EV1–3). A gradual and significant decline in alpha diversity was observed from HCs to BF1 to BF2 (Fig. 1A; Appendix Table S1). Beta diversity also differed significantly between HCs and both BF1 and BF2, while no significant difference was observed between BF1 and BF2 (Fig. 1B).

To identify the gut microbes specific to fibrosis based on their relative abundance, we constructed and compared four linear regression models (Guo et al, 2023), adjusting for age, sex, body mass index (BMI), smoking status, and disease location (only in CD patients). Model 1 compared CD patients with HCs; model 2 compared BF1 patients with HCs; model 3 compared BF2 patients with HCs; and, model 4 compared BF1 patients with BF2 patients. All analyses were restricted to the genus level. Model 1 identified 67 genera with differential abundance in CD patients compared to HCs (FDR < 0.1). These included HC-enriched *Alistipes*, *Roseburia* and *Faecalibacterium* and CD-enriched *Escherichia_Shigella*, *Ruminococcus__gnavus_group*, and *Streptococcus*; these findings were consistent with previous reports (Schirmer et al, 2018; Schirmer et al, 2019) (Dataset EV4). Model 2 identified 61 genera (Dataset EV5), while model 3 identified 71 genera (Dataset EV6). No significantly different microbes were observed in model 4. Nevertheless, seven genera were identified in model 3 that were not identified in models 1 or 2, indicating their specificity to BF2 rather than an enrichment in patients with CD (Fig. 1C; Table EV2, Appendix Table S1). Among these genera, the relative abundance of *Romboutsia*, *Lachnospiraceae_NK4A136_group*, *Ruminococcus_2*, *Ruminococcaceae_UCG_004*, *Muribaculaceae*, and *Anaerotruncus* was decreased, whereas that of *Mogibacterium* increased with fibrosis severity. Although most of these genera (e.g., *Romboutsia*, *Lachnospiraceae_NK4A136_group*, *Ruminococcaceae_UCG_004*, *Anaerotruncus*, and *Muribaculaceae*) have been reported to exhibit similar alterations in studies on fibrosis in other organs (Lee et al, 2020; Albhaisi et al, 2021; Lee et al, 2021; Yu et al, 2021), the relationship between these bacteria and intestinal fibrosis was revealed for the first time in this study.

Similar to the results regarding microbial genera, the differences in fecal and blood metabolites between the BF1 and BF2 groups were not as pronounced as those observed between CD patients and HCs. Nevertheless, linear regression models revealed 1 fecal and 12 blood metabolites that exhibited significant abundance specifically in BF2 patients after correcting for potential covariates (see "Methods") (FDR < 0.1; Fig. 1D,E; Tables EV3 and 4, Appendix Table S1). To our best knowledge, there have been no reports on the association between Xylose and intestinal fibrosis, requiring further investigation. Notably, glutamine levels in blood samples

exhibited a significant decrease specifically in the BF2 group. We therefore speculated that glutamine may have protected patients with CD from fibrotic complications, consistent with a previous study showing that glutamine supplementation prevented fibrosis development in rats (San-Miguel et al, 2010). Conversely, serine induces intestinal fibrosis, similar to its effect on pulmonary fibrosis, and may exert its profibrotic effects by inducing fibroblast proliferation (Bernard, 2018).

Collectively, these results indicated that the severity of intestinal fibrosis was associated with alterations in the gut ecosystem. We revealed both potentially harmful and beneficial microbial and metabolomic features, suggesting their potential for diagnosing CD behaviors and warranting further analysis.

## Omics-based machine learning models reveal biomarkers for characterizing intestinal fibrosis

Because the identified BF2-specific microbial and metabolomic signatures indicated the potential for noninvasive discrimination of intestinal fibrosis severity using these signatures, we next investigated the performance of six machine learning approaches in distinguishing between CD patients and HCs as well as between BF1 and BF2 patients. The tested approaches included decision tree, random forest, gradient-boosted decision tree, extra tree, logistic regression and support vector machine methods. Five models were incorporated in each analysis where the predictors were clinical features (age, sex, BMI, smoking, disease location), gut microbiota, fecal metabolites, blood metabolites, respectively, and all features combined. The first four models were defined as single molecular models, while the last model was a combined model. These analyses were performed on a cohort of 152 patients based on all of the obtained multiomics data.

Overall, all machine learning approaches exhibited satisfactory performance in distinguishing CD patients from HCs (Tables EV5 and EV6). For distinguishing BF2 patients from BF1 patients, the random forest and gradient-boosted decision tree methods were the most effective machine-learning approaches (Table EV7). Using the random forest method, blood metabolites were identified as the superior single molecular classifier for distinguishing BF2 patients from BF1 patients compared to clinical features, gut microbiota, and fecal metabolites, although the performance of the combined model surpassed that of any single molecular model (Fig. 2A; Table EV8). When comparing the selected features across all the machine learning models, they consistently showed good performance in at least five of the methods. Three genera (*Ruminococcaceae_UCG_003*, *Solobacterium*, *Erysipelatoclostridium*), two fecal metabolites (linoleylcarnitine and glycoursodeoxycholic acid (GUDCA)) and six blood metabolites (citramalic acid, PE.36:2, PE.36:1, PE.36:0, PE.34:2, DAG.36:5) were finally selected (Fig. 2B). Moreover, to assess the generalizability of the combined random forest model with these 11 predictors, we applied it to an external test cohort consisting of data from 64 patients with CD (BF1, $n = 20$; BF2, $n = 44$; Table EV1) using identical recruitment criteria as the derivation cohort and found that it also demonstrated robust performance in this cohort (Fig. 2C; Table EV9).

Taken together, we further identified several robust microbial and metabolomics features as potential biomarkers of moderate-severe intestinal fibrosis. We also provided a classifier (Computer

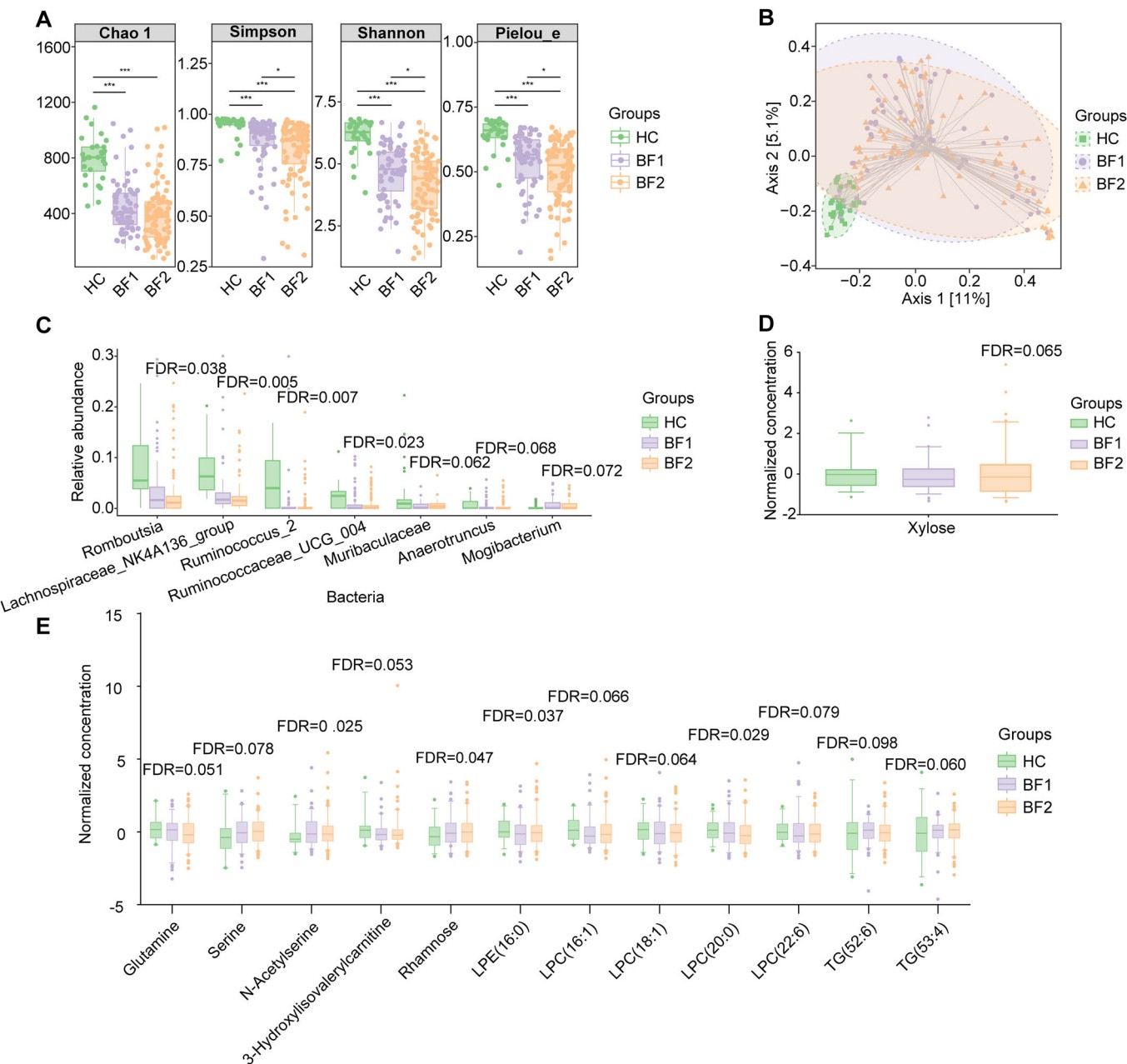

**Figure 1. The gut microbial and fecal/blood metabolomic alterations in patients with CD with various degrees of intestinal fibrosis.**

(A) Alpha diversity, measured by the Chao1, Simpson, Shannon and Pielou_e indices, is compared among the HCs, BF1 and BF2 groups ($n = 28$, 64, and 88 per group; $n$, sample size) by Kruskal–Wallis test ($P = 5.1e\text{-}12$, $P = 1.1e\text{-}09$, $P = 1.8e\text{-}11$ and $P = 4.6e\text{-}10$, respectively). *$P < 0.05$; ***$P < 0.001$. (B) Beta diversity represented by Bray–Curtis distances between the HCs, BF1 and BF2 groups ($n = 28$, 64, and 88 per group). The statistical method used for beta diversity is permutation ANOVA. Beta diversity differs significantly between the three groups (both $P = 0.001$), while no significant difference is observed between BF1 and BF2 groups ($P = 0.550$). (C) Seven BF2-specific bacterial genera are identified by four models (FDR < 0.1; Linear regression, $T$ test). (D, E) One BF2-specific fecal metabolite (D) and twelve BF2-specific blood metabolites (E) are identified (all FDR < 0.1; Linear regression, $T$ test). The FDR in charts (C–E) indicates the significance of the comparison between HCs ($n = 28$) and BF2 patients ($n = 88$, 88, 121 for charts (C–E), respectively). The Y axis of (C) indicates the bacterial relative abundance and Y axis of (D, E) indicates the normalized concentration using z-score standardization. Boxplots (A, C–E) represent the interquartile ranges (25th through 75th percentiles, boxes), medians (50th percentiles, bars within the boxes), and the 5th and 95th percentiles (whiskers below and above the boxes). The dots in plots (A, B) represent individual participants, while in plots (C–E) indicate the outliers. Source data are available online for this figure.

Code EV1) for distinguishing the severity of intestinal fibrosis in patients with CD, filling a gap in the lack of microbial and metabolomics tools in this field.

## Microbes and their metabolites are associated with fibrosis-related luminal and extraluminal morphological alterations

Intestinal fibrosis usually leads to luminal and extraluminal morphological alterations that can be clearly detected by MRE (an imaging tool widely used in clinical settings for evaluating bowel lesions and their response to treatment) (Bruining et al, 2018). Our effective delineation of the distinctive microecological characteristics of intestinal fibrosis prompted us to investigate the

potential correlations of the identified key gut microbes and metabolites with intestinal fibrosis-associated MRE features. To investigate these correlations, we first comprehensively evaluated the MRE findings of intestinal lesions (Bruining et al, 2018), regardless of whether or not they had previously been linked to fibrosis. These MRE features included eight luminal (i.e., stricture, penetration, wall thickening, mural edema, mural enhancement degree, mural enhancement pattern, length of diseased bowel, and mural apparent diffusion coefficient) and four extraluminal (i.e., perienteric effusion, comb sign, adenopathy, and perianal diseases) features (Table EV10). From this analysis, we identified four significant MRE features (stricture, penetration, effusion and comb sign) and conducted a meta-analysis, as BF1 and BF2 patients showed distinct multiomics patterns. During this analysis, age, sex,

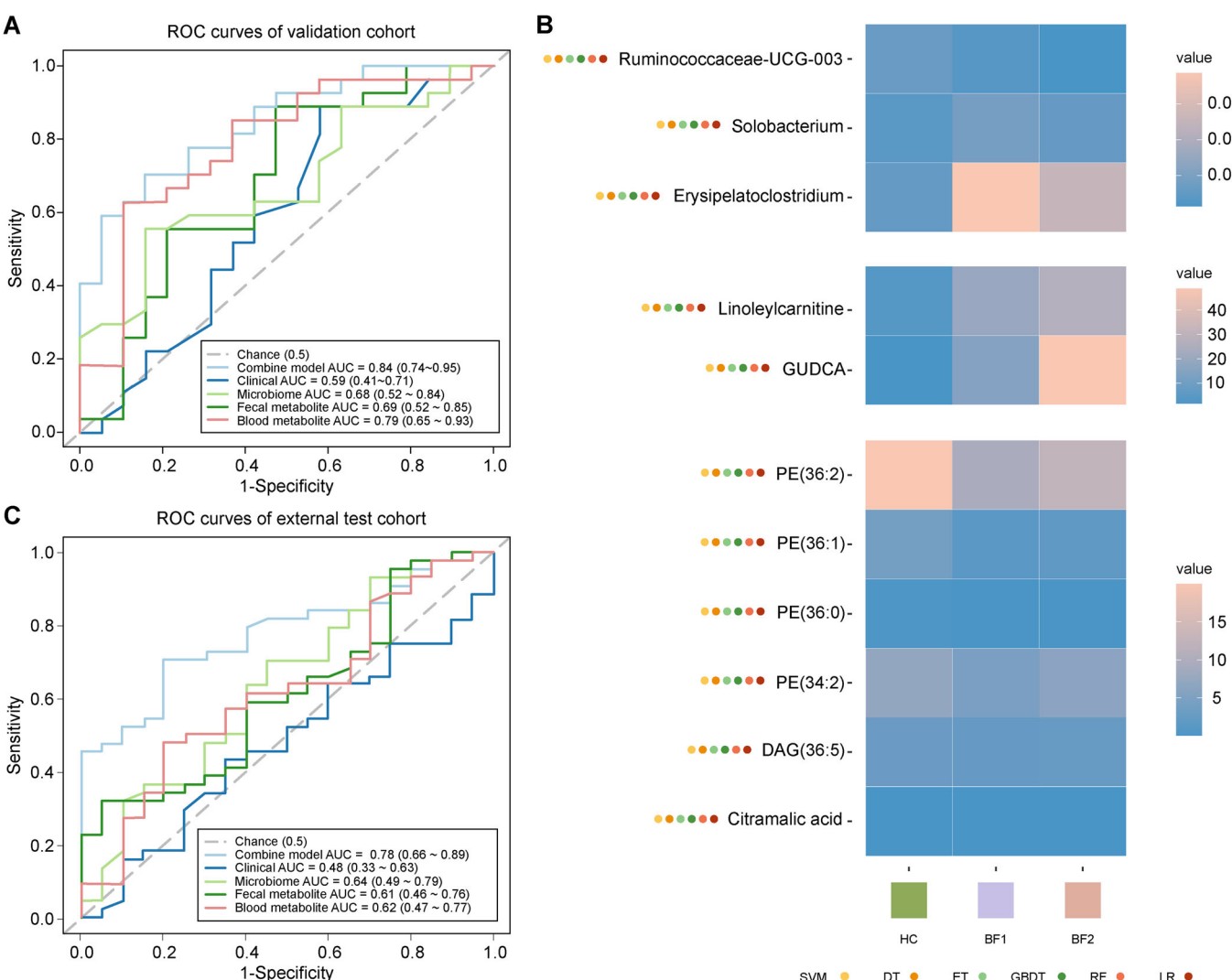

**Figure 2. Prediction models based on integrating omics data layers.**

(A) AUC plots of the four single molecular models and the combined model for the validation cohort. The combined model shows the best diagnostic performance for intestinal fibrosis using the random forest algorithm, with an AUC of 0.84 (95% CI: 0.73–0.95). (B) Microbial and metabolomic features selected by six different machine learning-based models (SVM support vector machine, DT decision tree, ET extra tree, GBDT gradient-boosted decision trees, RF random forest, LR logistic regression). Darker pink color indicates greater feature importance. (C) AUC plots of the four single molecular models and the combined model in the external test dataset. The combined model shows the best performance using the random forest algorithm, with an AUC of 0.78 (95% CI: 0.66–0.89). Source data are available online for this figure.

BMI, smoking status and disease location were corrected to avoid potential confounding effects. In total, 215 significant signals that presented consistent molecular trait-MRE correlations across BF1 and BF2 patients were identified (FDR < 0.1, heterogeneity test P value > 0.05), including nine bacterial associations, 20 fecal metabolic associations and 186 blood metabolic associations (Fig. 3A–D; Dataset EV7).

Among the four significant MRE features, the severity of bowel stricture and penetration were correlated with changed abundance of the genera *Escherichia_Shigella*, *Saccharimonadales* and *Muribaculaceae* (Fig. 3B). The identification of *Muribaculaceae* in this case overlapped with our previous results from the analysis of microbiota specific to BF2 patients. Stricture and penetration are the imaging phenotypes most associated with intestinal fibrosis and lead to surgical resection and significant healthcare costs (D'Alessio et al, 2022). Therefore, they are a top concern for clinicians treating patients with CD, highlighting the urgent need to explore these relationships. A cohort study of pediatric CD patients implicated *Ruminococcus* in stricture and *Veillonella* in penetrating diseases (Kugathasan et al, 2017). Here, we revealed a number of gut microorganisms associated with intestinal stenosis and penetration that might be specific to adult patients with CD. Furthermore, blood and fecal metabolites showed more associations with these two imaging phenotypes than did the gut microbiota. Notably, the level of microbiota-derived indole-3-propionic acid (IPA) in blood was decreased in the severe stricture group (Dataset EV7). The administration of IPA has been reported to strengthen intestinal homeostasis through IL-10 signaling (Alexeev et al, 2018) or shape the intestinal mesenchyme via pregnane X receptor signaling (Flannigan et al, 2023), making it a promising new therapy for intestinal inflammation and fibrosis.

The other two significant MRE features, namely, perienteric effusion and comb sign, which reflect the severity of the inflammation of mesenteric adipose tissue around the diseased gut, also presented a broad range of associations with metabolites and the gut microbiota. For example, the level of docosahexaenoic acid (DHA), an important omega-3 polyunsaturated fatty acid (PUFA), was decreased in patients with both severe effusion and comb sign (Dataset EV7). Omega-3 PUFAs are essential fatty acids obtained from the human diet that benefit gut immunity and homeostasis (Gentile and Weir, 2018). The establishment of this "distance" relationship between the luminal microbiota and extraluminal imaging signs may be due to the translocation of the gut microbiota to mesenteric adipose tissue driving the pathogenic expansion of this tissue (Ha et al, 2020). This, in turn, promotes the development of intestinal fibrosis (Li et al, 2021; Qian et al, 2023). These results once again highlighted the intricate pathogenic mechanism underlying intestinal fibrosis, involving a complex interplay of luminal, transmural, and extraluminal factors.

The associations between molecular traits and MRE features exerted only in BF1 or BF2 group (heterogeneity test P value < 0.05) (Dataset EV8). In particular, certain blood amino acids and their derivatives, such as *L*-aspartic acid, 4-hydroxyproline and methylcysteine, were significantly associated with MRE features only in BF2 patients and not in BF1 patients (Fig. 3E–H). Moreover, the short-chain fatty acid, namely ethylmethylacetic acid (a-methyl butyrate), exhibited a negative correlation with the penetration sign in BF2 patients, suggesting its potential as a protective factor against disease progression (Ma et al, 2023).

Taken together, these findings suggested that the gut microbiota, along with fecal and blood metabolites, may contribute to both luminal and extraluminal morphological changes associated with intestinal fibrosis. However, some of these associations were stronger in patients with BF2, indicating a fibrosis degree-dependent effect on relationships between microecological characteristics and MRE features.

## Microbial metabolites as potential key actors in intestinal fibrosis

Since fecal metabolites have been shown to serve as functional readouts of the gut microbiota, we conducted mediation analysis to establish putative causal links from the gut microbiota to fibrosis-associated morphological changes mediated by small molecules. Briefly, four models in which bacteria were considered the exposure, MRE features the outcome and blood or fecal metabolites the potential mediators were used to assess the data of BF1 and BF2 patients (see "Methods").

Not surprisingly, a larger number of significant mediation effects were observed among fecal molecules than among blood molecules (FDR$_{ACME}$ < 0.1; Figs. 4A,B and EV2A,B; Datasets EV9–12). Strikingly, a major group of genera belonging to Lachnospiraceae, including *Lactobacillus*, *Roseburia*, *Marvinbryantia*, *Eubacterium ventriosum* and *Eubacterium hallii*, showed potential protective effects against stricture, penetration and perienteric effusion in BF2 patients through fecal metabolites (Dataset EV10). All of these bacteria have been previously reported to be beneficial to intestinal function (Lavelle and Sokol, 2020). We also identified an anti-inflammatory immunonutrient (glycine) that is partially derived from food digestion by the gut microbiota as a putative mediator of the negative associations between Lachnospiraceae groups and intestinal stricture (Fig. 4A).

Importantly, when we examined the overlap between the small molecules identified in blood and fecal samples, we observed that *L*-aspartic acid and its derivatives (e.g., N-acetyl aspartic acid), which might be metabolized by *Granulicatella*, *Dubosiella* and *Saccharimonadales*, were positively associated with the presence of stricture, penetration and perienteric effusion based on a more lenient threshold (all $P_{ACME}$ < 0.05) (Fig. 4C; Tables Datasets EV9–12). Moreover, metagenomics sequencing of a subset of fecal samples obtained from 71 participants (including 47 CD patients and 24 HCs randomly selected from the cohorts in this study) showed that the predicted microbial pathway was negatively associated with blood aspartic acid levels (alanine, aspartate and glutamate metabolism pathways [KO00250], $P = 0.022$, $r = -0.32$); We also observed a similar negative association between these two in fecal samples, although without statistical significance ($P = 0.400$, $r = -0.12$) (Fig. 4D,E). These findings suggested that *L*-aspartic acid, which was associated with microbial metabolic activities, may play a pivotal pathogenic role in the luminal and extraluminal morphological alterations. We performed additional analyses, including a comparison of *L*-aspartic acid levels among different inflammatory groups and a multiple linear regression analysis, to confirm that this relationship between *L*-aspartic acid and intestinal fibrosis was not significantly biased by coexisting intestinal inflammation (see "Methods"; Fig. EV3). *L*-aspartic acid is a nonessential amino acid that serves as a substrate for a variety of biosynthesis pathways, including essential amino acids and

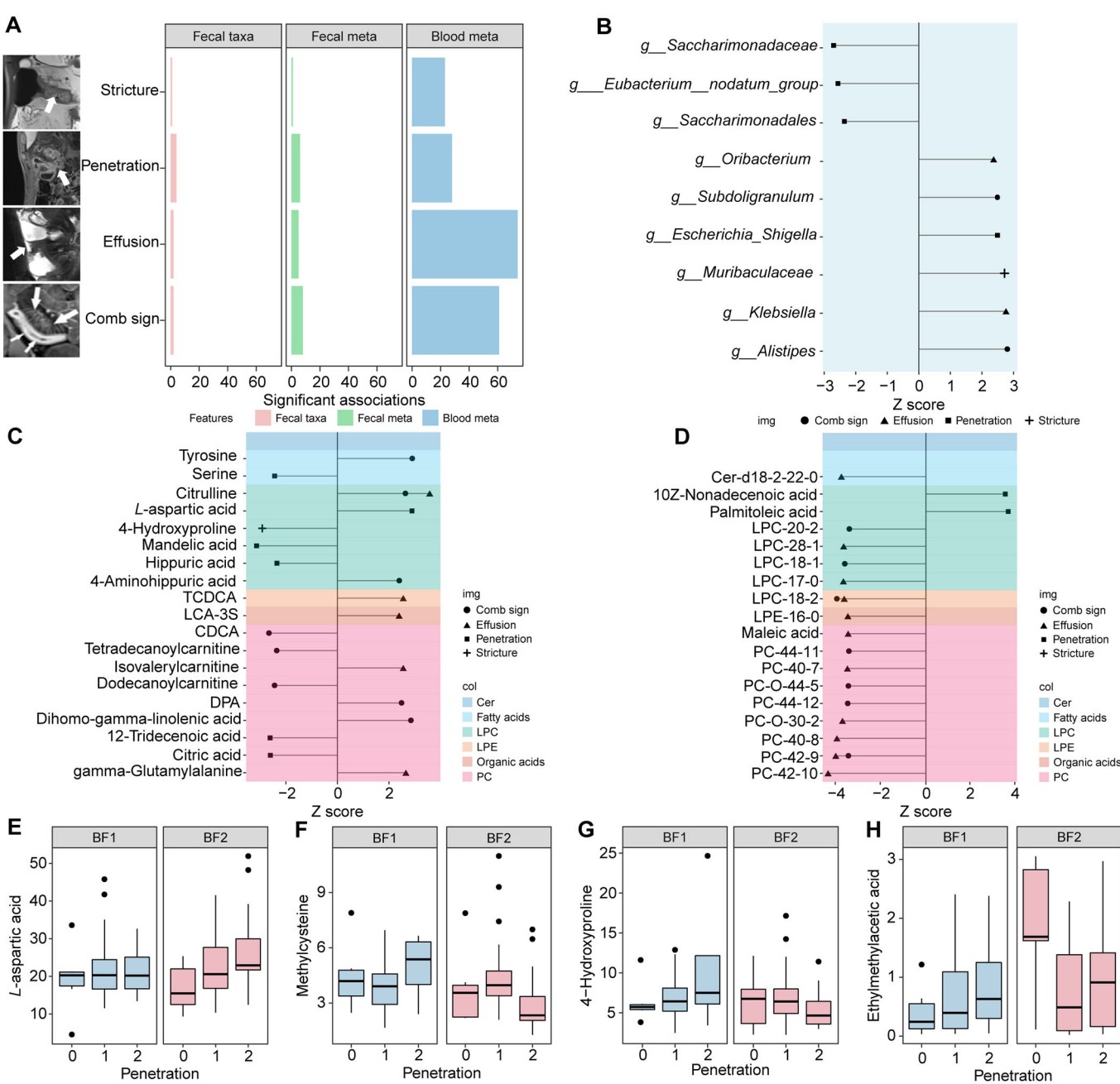

**Figure 3. Associations between MRE characteristics and omics features.**

(A) The total numbers of significant associations of taxa ($n = 9$), fecal metabolites ($n = 20$) and blood metabolites ($n = 186$) with four intestinal fibrosis-related MRE features (stricture, penetration, perienteric effusion and the comb sign) (FDR < 0.1). The arrows on the MRE images depicted along the left side of this panel indicate the corresponding image features. (B–D) Bubble plots indicating the different types of omics data associated with MRE that are consistent in BF1 and BF2 patients (meta FDR < 0.1, heterogeneity test $P$ value > 0.05), arranged from left to right, including gut microbiota members (B), fecal metabolites (C) and blood metabolites (D). The $X$ axis indicates the $Z$ value from meta-analysis, while the $Y$ axis represents the features. Exact data are presented in Dataset EV7. (E–H) Four examples of associations between specific blood metabolites and penetration with different patterns in BF1 ($n = 64$) and BF2 ($n = 88$) patients (heterogeneity test $P$ value < 0.05), including $L$-aspartic acid (E), methylcysteine (F), 4-hydroxyproline (G) and ethymethylacetic acid (H). The $X$ axis indicates the degree of intestinal penetration detected with MRE, while the $Y$ axis represents the normalized abundance values of metabolites. Exact data are presented in Dataset EV8. Boxplots (E–H) represent the interquartile ranges (25th through 75th percentiles, boxes), medians (50th percentiles, bars within the boxes), and the 5th and 95th percentiles (whiskers below and above the boxes). The dots in plots (E–H) indicate the outliers. Source data are available online for this figure.

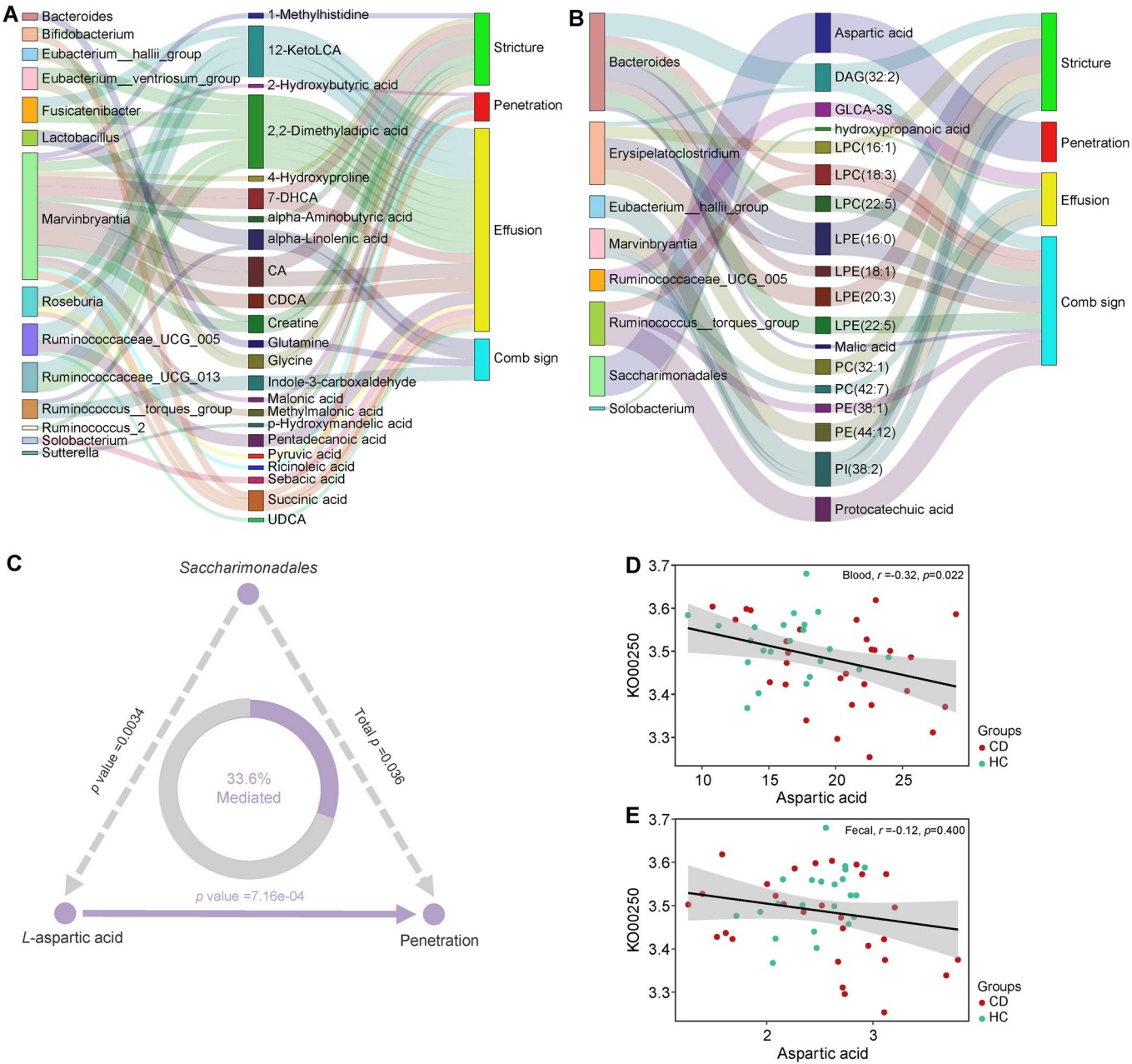

**Figure 4. Putative links between the gut microbiota, metabolites and MRE characteristics.**

(A, B) Sankey plots indicating bacterial contributors to MRE features mediated by fecal metabolites ($n = 23$; **A**) and blood metabolites ($n = 44$; **B**) in BF2 patients by mediation analysis. The remarkable link is L-aspartic acid, which might be metabolized by *Saccharimonadales* and is positively associated with the presence of intestinal penetration. All P values of average causal mediation effects <0.05. Color boxes represent the gut microbiota (left column), metabolites (intermediate column) and MRE features (right column), respectively. The line width represents the relative size of P value for the correlation between the two "boxes" (wider lines indicate relatively smaller P values). (**C**) Illustration of the association between *Saccharimonadales* and MRE-detectable intestinal penetration mediated by L-aspartic acid in BF2 patients, as an example ($P_{ACME} = 0.008$; Dataset EV12). (**D, E**) Alanine, aspartate and glutamate metabolism [KO00250] predicted from metagenomics sequencing data. The associations between the abundance of KO00250 and aspartic acid levels in blood ($r = -0.32$, $P = 0.022$; **D**) and feces ($r = -0.12$, $P = 0.400$; **E**) are shown using the Spearman correlation test. Source data are available online for this figure.

enzymes (Caspi et al, 2020). Elevated levels of L-aspartic acid have been detected in the blood of patients with liver fibrosis (Vannucchi et al, 1985), and it has been reported to be a crucial factor in driving myocardial fibrosis (Ritterhoff et al, 2020). However, its role in intestinal fibrosis remains unclear.

Based on all of these results, we proposed potential connections whereby microbiota-derived metabolites induce intestinal fibrosis and subsequently result in MRE-detectable intestinal morphological alterations. Among these links, further investigation into the role of L-aspartic acid in promoting intestinal fibrosis is warranted,

as it has been implicated in driving fibrosis in the liver and heart (Vannucchi et al, 1985; Ritterhoff et al, 2020).

## *L*-aspartic acid promotes the progression of intestinal fibrosis in vitro and in vivo

To investigate the modulatory effect of *L*-aspartic acid on intestinal fibrosis, we compared the progression of intestinal fibrosis in rats (induced by trinitrobenzene sulfonic acid [TNBS] administration) with or without *L*-aspartic acid treatment, while using rats treated with phosphate buffer saline (PBS) as healthy controls. The TNBS-induced rat model is widely used to study CD due to its ability to rapidly induce colonic inflammation and subsequent intestinal fibrosis (Dillman et al, 2013). We also used MTI and $^{18}$F-fibroblast activation protein inhibitor (FAPI) PET/CT (another novel imaging technique for noninvasive assessment of intestinal fibrosis by targeting fibroblast activation protein (Scharitzer et al, 2023) to observe the intestines of rats. Overall, the degree of intestinal disorder found in TNBS-induced rats was significantly higher after three weeks of *L*-aspartic acid treatment than that in the control groups, in line with our hypothesis. Specifically, the more severe intestinal morphological changes observed that the rats in the treated group compared with control group 1 (stimulated by TNBS only) included the development of intestinal penetration and adhesion, observed both via imaging and in gross specimens (Fig. 5A). The uptake of $^{18}$F-FAPI in intestinal lesions in the treated group (treated with TNBS and *L*-aspartic acid) was significantly higher than that in either control group 1 or control group 0 (no TNBS or *L*-aspartic acid treatment), indicating an increased degree of intestinal fibrosis in the treated group (Fig. 5B; Appendix Table S2). The fibrosis-promoting effect of *L*-aspartic acid was also demonstrated by a significant increase in the normalized magnetization transfer ratio (MTR) observed in the treated group, as compared to both control groups 0 and 1 (Fig. 5C; Appendix Table S2). Masson trichrome staining was further conducted to evaluate intestinal fibrosis, and the pathological findings were clearly consistent with all imaging results. Specifically, the fibrotic scores of the treated group were higher than those of control groups 0 and 1 (Fig. 5D; Appendix Table S2). These findings provided direct evidence of the aforementioned hypothesized correlations between small molecules and macroscopic morphology in intestinal fibrosis.

Moreover, we investigated the impact of *L*-aspartic acid on intestinal fibrosis function in human intestinal myofibroblasts (HIMFs) which are the major effector cells of intestinal fibrosis. HIMFs were cocultured with TGF-β1 (which are markers indicating activation of HIMFs) and exposed to different concentrations *L*-aspartic acid (0, 6.5 or 13 μM); a control group without any treatment was also included. Consistent with our predictions, the mRNA expression of collagen 1A1 (*COL1A1*), fibronectin (*FN1*), and actin alpha 2 (*ACTA2*) in HIMFs was markedly increased following treatment with TGF-β1 compared to the control group, and coculture with *L*-aspartic acid further amplified this effect in a dose-dependent manner (Fig. 6A–C; Appendix Table S3). The profibrotic effect of *L*-aspartic acid was also observed at the protein level, as evidenced by a significant upregulation of the expression of COL1A1 and FN under treatment with this amino acid (Fig. 6D–F; Appendix Table S3). These data suggested that *L*-aspartic acid might act synergistically with the TGF-β1-induced fibrogenic activation of HIMFs by upregulating the extracellular matrix at both the mRNA and protein levels.

Taken together, these results demonstrated that *L*-aspartic acid promoted the progression of intestinal fibrosis both in vitro and in vivo, providing evidence of a causal relationship between this specific metabolite (or the gut microbiota that produce it) and intestinal fibrosis.

## Discussion

Although previous animal studies have linked the gut microbiota or metabolites to intestinal fibrosis, specific microbial and metabolite targets related to intestinal fibrosis in patients with CD have remained unclear. The main constraint on the implementation of such cohort studies is the lack of effective tools for quantifying transmural fibrosis. Here, we utilized MTI, a proven imaging tool, to achieve the accurate classification of intestinal fibrosis in patients with CD. This enabled us to present, for the first time, a series of microbial and metabolite profiles that are specific to moderate-severe intestinal fibrosis in patients with CD. A further innovation of our study was the establishment of a "bridge" between the small molecule profiles and the macroscopic imaging phenotypes of intestinal fibrosis, and the results emphasized the potential to target these traits identified by multiomics analysis to effectively treat fibrosis. Subsequently, we selected *L*-aspartic acid as an example among numerous targets and obtained causal evidence that it induces fibrotic gene activation in HIMFs in vitro and promotes the aggravation of intestinal fibrosis in rats in vivo. This further indicates the efficacy of targeting these specific molecules to treat intestinal fibrosis.

CD is associated with alterations in the gut microbiota and metabolites, whose potential involvement in the development of intestinal fibrosis is currently under investigation in both human and mouse studies. For example, fibrosis-triggering gut microbes have been identified in the cecum (the species *Mucispirillum schaedleri* and the genera *Ruminococcus* and *Anaeroplasma*) and ileum (the genera *Streptococcus* and *Lactobacillus*) in an animal study (Jacob et al, 2018). The colonization of adherent-invasive *Escherichia coli* (AIEC) in the gut has been reported to be associated with severe intestinal fibrosis in eight patients with CD (Xu et al, 2023) and in mice (Small et al, 2013; Ellermann et al, 2019). Fecal and blood metabolites can serve as functional readouts of links between the gut microbiota and intestinal fibrosis. In a human intestinal organoid model, eicosatetraynoic acid and butyrate were found to regulate extracellular matrix genes implicated in stricture and to suppress collagen accumulation and tissue stiffness (Jurickova et al, 2022). Although there has been progress in identifying microbes and metabolites associated with fibrosis in CD, a significant gap still exists between the current state and clinical implementation due to the lack of extensive patient analysis. Transcending previous studies, we report a relatively complete landscape of the gut microbiota associated with moderate-severe intestinal fibrosis in patients with CD. Notably, a significant decrease of alpha diversity was identified in BF2 compared with BF1, but no significantly changed taxa identified when comparing BF2 with BF1 group. One possible explanation was the limited sample size and another hypothesis was that individual bacteria (e.g., *Ruminococcaceae*, *Muribaculaceae*) and

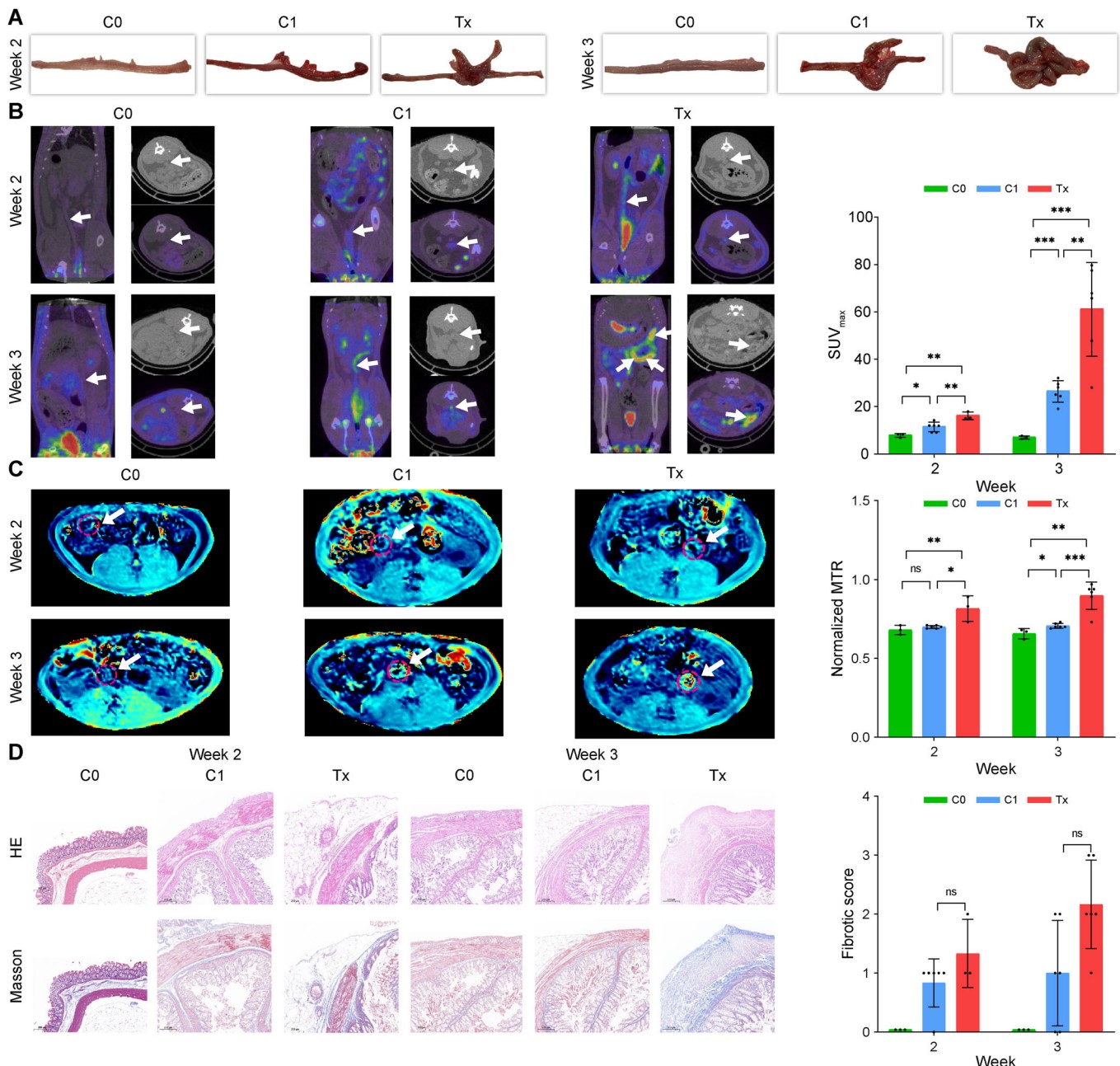

**Figure 5. L-aspartic acid promotes the progression of intestinal fibrosis in vivo.**

(A) Colorectal segments of rats in the Tx (2 weeks, n = 3; 3 weeks, n = 6), C1 (2 weeks, n = 6; 3 weeks, n = 6) and C0 (2 weeks, n = 3; 3 weeks, n = 3) groups. The colorectums of the Tx group rats exhibit more severe morphological alterations, including penetration and adhesion, than those of the C1 and C0 group rats. (B) [18]F-FAPI PET/CT imaging demonstrates significantly higher FAPI uptake in rats in the Tx group than in those in the C1 and C0 groups (all P < 0.05), suggesting a profibrotic effect of L-aspartic acid. The white arrows point to normal intestine (C0 group) or intestinal lesions (Tx and C1 groups). The left image in each panel represents a coronal PET/CT scan, while the right images display axial CT (top right) and PET/CT (bottom right) scans at the same level. (C) MTI also demonstrates a significantly higher normalized MTR in rats in the Tx group than in those in the C1 and C0 groups (all P < 0.05), providing further evidence of the profibrotic effect of L-aspartic acid. The pink circle denotes the location of the colorectum. (D) Hematoxylin–eosin (HE) and Masson staining of intestinal specimens from rats in the Tx group shows higher collagen deposition than in those from the C1 and C0 groups, although this difference does not reach statistical significance. However, this observation is consistent with the findings observed through [18]F-FAPI PET/CT and MTI. C0 represents the control group without treatment with TNBS or L-aspartic acid, while C1 indicates the group treated with TNBS, and Tx denotes the group receiving additional administration of L-aspartic acid under TNBS treatment. All data are presented as the mean ± SD. The comparative analysis of FAPI uptake and MTR among the C0, C1, and Tx groups is performed using the T test. The pathological findings (HE and Masson staining) among the three groups are analyzed using the Fisher's exact test. *P < 0.05; **P < 0.01; ***P < 0.001; ns not significant. The dots in the plots indicate individual rats. Source data are available online for this figure.

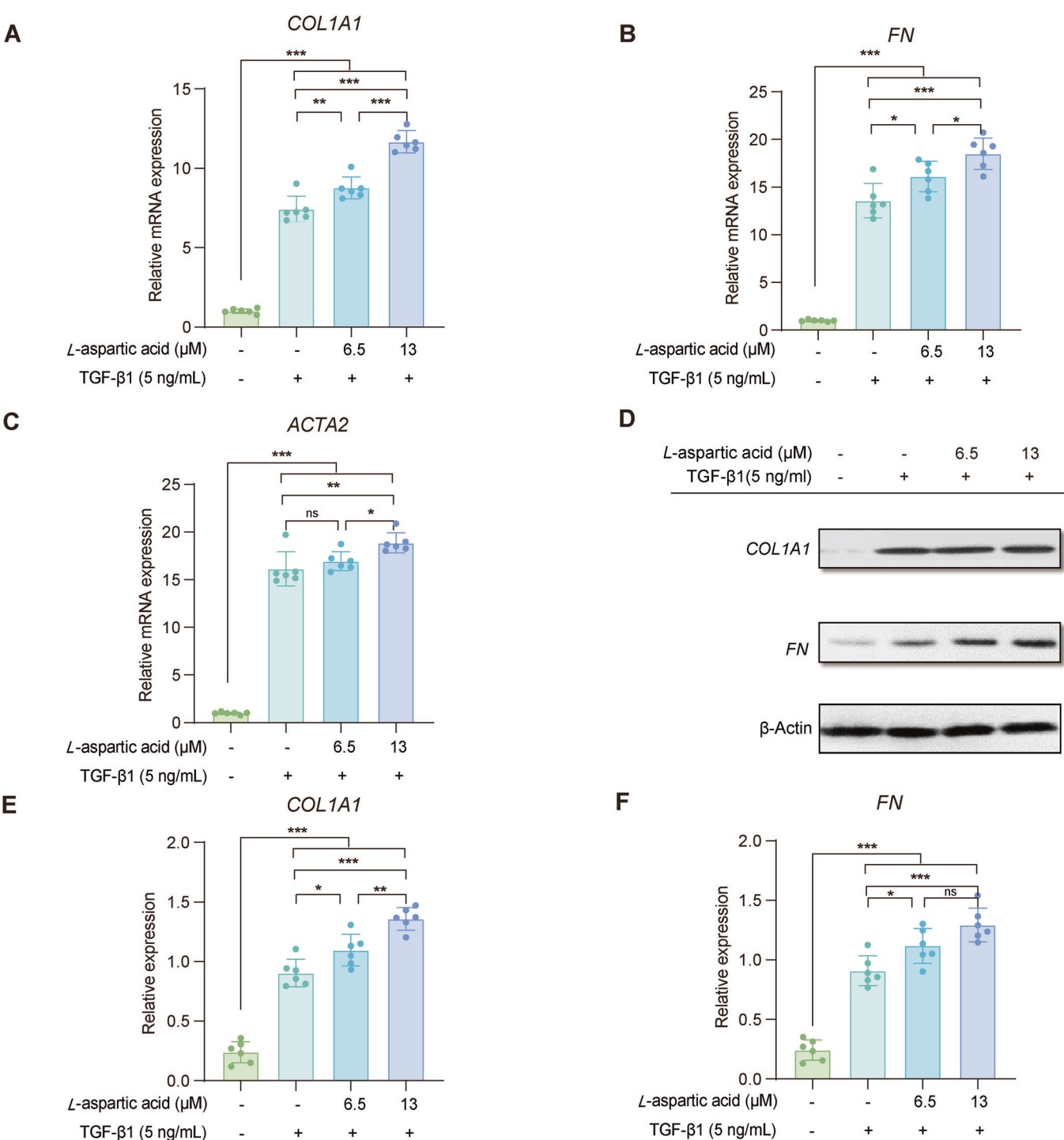

**Figure 6. *L*-aspartic acid promotes the progression of intestinal fibrosis in vitro.**

*L*-aspartic acid acts synergistically with TGF-β1-induced fibrogenic activation of human intestinal myofibroblasts (HIMFs). HIMFs were treated with TGF-β1 (5 ng/mL) and cocultured with or without *L*-aspartic acid. (A–C) Results of qPCR analysis of the relative mRNA expression of collagen 1A1 (*COL1A1*), fibronectin (*FN1*), and actin alpha 2 (*ACTA2*). The data are normalized to β-actin expression and are presented as relative values compared with the control ($n = 6$ for each group). Cells were seeded at $1.5 \times 10^5$ cells/well. (D) Representative western blots showing the protein expression of Procol1A1 and FN with β-actin as a loading control. (E, F) Quantitation of COL1A1 and FN from western blot analyses ($n = 6$ for each group). Both qPCR analysis and western blots has been biologically replicated in three HIMFs and technically repeated for six times per HIMF. Data are expressed as the mean ± SD of six technical replicates of the same HIMF. Statistical significance is determined by one-way ANOVA analysis with Tukey's post hoc test. *$P < 0.05$; **$P < 0.01$; ***$P < 0.001$; ns not significant. Bars represent Standard Deviation. The dots in the plots indicate individual wells. Source data are available online for this figure.

fecal/blood metabolites (e.g., *L*-aspartic acid, glutamine, IPA) might be specific to BF2 in a non-linear relationship along with the severity of fibrosis, which was consistent with some findings reported in a previous meta-analysis (Ma et al, 2022). These biomarkers were derived from the largest patient cohort with CD related to this research topic reported to date and were externally validated with a test cohort. Among these factors, microbial metabolites, particularly amino acids, appeared to have a greater impact on the progression of intestinal fibrosis than the gut microbiota itself. The metabolism of *L*-aspartic acid by gut microbes was supported by our metagenomics sequencing data. Although the involvement of *L*-aspartic acid has been documented in other types of organ fibrosis (Vannucchi et al, 1985; Ritterhoff et al, 2020), our study is the first to demonstrate its role in intestinal fibrosis. *L*-aspartic acid-induced activation of the profibrotic phenotype of fibrotic genes in HIMFs and promoted aggravation of intestinal fibrosis in an animal model. We speculate that specifically inhibiting the production or uptake of *L*-aspartic acid in the fibrotic intestine through targeted drugs or dietary interventions has the potential to effectively halt the progression of intestinal fibrosis. Thus, at least one potential target that we identified holds promise for advancing intestinal fibrosis research.

Notably, these biomarkers were selected based on the MTI of the intestines rather than the Montreal classification of patients with CD. The image contrast in MTI depends on the amount of macromolecules such as collagens in intestinal tissue. The normalized MTR can be used to measure this MT effect and reflects the concentration of macromolecules in an aqueous physiological environment (Li et al, 2018). Our work has consistently demonstrated a strong correlation between MTI and pathological fibrosis (Li et al, 2018; Fang et al, 2020; Meng et al, 2020; Lu et al, 2021), which is supported by similar conclusions reached at other medical institutions (Adler et al, 2011; Adler et al, 2013; Pazahr et al, 2013; Dillman et al, 2015). MTI is a reliable imaging technique that accurately and noninvasively assesses transmural fibrosis without being affected by coexisting inflammation (Li et al, 2018), whereas Montreal classification, such as B2 and B3, are clinical phenotypes resulting from the combined effects of inflammation and fibrosis. Therefore, these selected biomarkers are more specific for diagnosing and treating intestinal fibrosis in CD patients. Based on these intestinal microbiota and metabolite biomarkers, we developed and validated the first microecological model capable of accurately diagnosing intestinal fibrosis. This alternative diagnostic tool may be particularly useful in medical centers lacking advanced imaging technology such as MTI.

More intriguingly, our study also revealed relationships between the gut microbiota, metabolites, and intestinal morphological alterations. Such fibrosis-related intestinal morphological changes, particularly stricture and penetration, frequently cause severe clinical symptoms such as abdominal pain, nausea or vomiting and are therefore the primary focus of clinical management (Rieder et al, 2017). We speculated that the gut microbiota and/or its metabolites activated pathogen-associated molecular patterns via pattern recognition receptors, thereby inducing innate and adaptive immune responses (Takeuchi and Akira, 2010); by amplifying and sustaining these immune and inflammatory processes, intestinal fibrosis and disease complications such as stricture, penetration, and extraluminal manifestations would be promoted (Lee and Chang, 2021). This hypothesis was partially supported by our

in vivo and in vitro results, particularly by the more pronounced alterations in intestinal morphology observed in rats treated with *L*-aspartic acid compared with controls. Although this was not our primary objective, this study presents fairly comprehensive evidence that MRE-detectable intestinal macromorphological alterations are at least partially attributable to microbial and metabolite factors and provides novel insights into the potential mechanisms underlying these changes. More importantly, from a clinical imaging perspective, we confirmed that the selected microbes and metabolites could serve as reliable biomarkers for intestinal diagnosis and treatment based on our "bridge" connecting small molecules to macroscopic morphological changes.

There were several limitations in this study. First, we did not incorporate additional potential confounding factors such as patient dietary patterns and lifestyles during data analysis. The inclusion of these confounding factors, along with data collection from more centers, could strengthen our conclusions. Second, as the data suggested, we observe negative correlations between KO00250 and aspartic acid concentrations in both fecal and blood samples, indicating that the gut microbiota processed a metabolizing functionality of aspartic acid. However, the association between KO00250 and aspartic acid concentration in fecal samples was not statistically significant. The inconsistent relationships between blood and fecal metabolites have been repeatedly reported in previous studies (Chen et al, 2020; Deng et al, 2023), probably due to the difference of compound transformation, metabolic absorption, and chemical format. Further study is warranted to investigate the specific microbial taxonomy and gene functions which are involved in the aspartic acid metabolism process. Third, although we had provided evidence to support the role of *L*-aspartic acid in promoting intestinal fibrosis, further experiments are necessary to elucidate its specific mechanisms of action. Moreover, additional investigation should focus on the "gut microbiota–metabolites–intestinal fibrosis–intestinal morphological changes" axis to better understand its pathogenic mechanism.

In conclusion, we present a proof-of-concept study to support the development of microbial/metabolite-based tools for managing intestinal fibrosis in patients with CD. These findings provide promising potential therapeutic targets for CD-associated intestinal fibrosis. The subsequent development of specific drug treatment strategies or dietary interventions based on these targets may potentially offer a broader spectrum of therapeutic options for the management of intestinal fibrosis.

## Methods

**Reagents and tools table**

| Reagent/resource | Reference or source | Identifier or catalog number |
|---|---|---|
| **Experimental models** | | |
| Primary human intestinal myofibroblasts | Musso et al, 1999; Zhao et al, 2020 | N/A |
| **Antibodies** | | |
| COL1A1 (E8F4L) XP® Rabbit mAb | Cell Signaling | #72026 |

| Reagent/resource | Reference or source | Identifier or catalog number |
|---|---|---|
| Anti-Fibronectin primary antibody | Abcam | ab268020 |
| Anti-β-actin primary antibody | Proteintech | 66008-1-Ig |
| Goat anti-Rabbit IgG (H + L) Cross-Adsorbed Secondary Antibody | Invitrogen | A11008 |
| Goat anti-Mouse IgG (H + L) Highly Cross-Adsorbed Secondary Antibody | Invitrogen | A32723 |
| **Chemicals, enzymes, and other reagents** | | |
| MagBeads FastDNA Kit for Soil | MP Biomedicals, CA, USA | 116564384 |
| TNBS | Meilunbio | MB5547-1 |
| Olive oil | Solarbio | O8320 |
| Saline (0.9%) | China Otsuka Pharmaceutical Co., Ltd. | COPKB01-1.2306. |
| Dulbecco's minimal essential medium | GIBCO | C11965500BT |
| Fetal bovine serum | GIBCO | 10091148 |
| Antibiotic-Antimycotic | GIBCO | 15240062 |
| L-aspartic acid | MACKLIN | L800690 |
| Recombinant Human TGF-β1 | Peprotech | AF-100-21C |
| TRIzol reagent | Ambion | 15596018 |
| ReverTra Ace qPCR RT Master Mix Kit | TOYOBO | FSQ-201 |
| FS Essential DNA Probes Master | Roche | 6924492001 |
| RIPA buffer | Cell Signaling | 9806S |
| Protease and phosphatase inhibitors | Cell Signaling | 5872S |
| Blue loading Buffer | Cell Signaling | 7722S |
| **Software** | | |
| QIIME2 2019.4 | https://docs.qiime2.org/2019.4/tutorials/ | N/A |
| Cutadapt (version v1.1) | https://cutadapt.readthedocs.io/en/stable/ | N/A |
| map2slim | www.metacpan.org | N/A |
| Kraken2 | https://ccb.jhu.edu/software/kraken2/ | N/A |
| MMseqs2 | https://github.com/soedinglab/MMseqs2 | N/A |
| Megahit (v1.1.2) | https://github.com/voutcn/megahit | N/A |
| KOBAS | http://bioinfo.org/kobas/ | N/A |
| EggNOG | http://eggnogdb.embl.de/ | N/A |
| CAZy | http://www.cazy.org/ | N/A |
| Prism statistics software | GraphPad | N/A |
| R (4.2.1) | https://www.r-project.org/ | N/A |
| Python (2.7.11) | https://www.python.org/ | N/A |
| FastQC (version 0.11.7) | http://www.bioinformatics.babraham.ac.uk/projects/fastqc/ | N/A |
| Metaphlan 3 | https://github.com/biobakery/MetaPhlAn | N/A |
| Humann 2 | https://github.com/biobakery/humann | N/A |
| Trimmomatic (0.33) | http://www.usadellab.org/cms/index.php?page=trimmomatic | N/A |
| DADA2 (1.03) | https://benjjneb.github.io/dada2/ | N/A |
| **Other** | | |
| NanoDrop NC2000 spectrophotometer | Thermo Fisher Scientific, Waltham, MA, USA | N/A |
| Vazyme VAHTS DNA Clean Beads | Vazyme, Nanjing, China | N/A |
| Quant-iT PicoGreen dsDNA Assay Kit | Invitrogen, Carlsbad, CA, USA | N/A |
| Illumina NovaSeq 6000 | Illumina | N/A |
| Qubit™ 4 Fluorometer | Invitrogen, USA | N/A |
| TruSeq Nano DNA High Throughput Library Prep Kit (96 samples) | Illumina, USA | 20015965 |
| N-(3-(dimethylamino)propyl)-N′-ethylcarbodiimide (EDC)·HCl | Sigma-Aldrich, St. Louis, MO, USA | N/A |
| ultraperformance liquid chromatography coupled to tandem mass spectrometry (UPLC–MS/MS) | ACQUITY UPLC Xevo TQ-S, Waters Corp., Milford, MA, USA | N/A |
| Targeted Metabolome Batch Quantification (TMBQ) | Metabo-Profile, Shanghai, China | v1.0 |
| iMAP platform | Metabo-Profile, Shanghai, China | v1.0 |
| Hematoxylin–eosin | Servicebio | G1076 |
| Masson | Servicebio | G1006 |
| Roche Light Cycler 96 instrument | Roche | 05815916001 |
| PAGE Gel Fast Preparation Kit | EpiZyme | PG112 |
| PVDF membrane | Invitrogen | LC2007 |

## Ethics

The human study protocols were approved by the Institutional Ethics Review Board of our hospital (No. [2021]215-2). Informed consent was obtained from all participants, and the experiments conformed to the principles set out in the WMA Declaration of Helsinki and the Department of Health and Human Services Belmont Report. The methods were performed in accordance with STROBE guidelines. The animal experiments in this study were approved and performed in accordance with the guidelines of the Animal Ethics Committee of our hospital (No. [2023]116). Animals were provided with ad libitum access to a standard laboratory diet and sterile water and were housed under controlled, monitored temperature and humidity conditions, in rooms with a 12 h light/dark cycle.

## Human participants

From May 2021 to March 2023, 291 patients with CD presenting for fecal and/or serum sample collection within one week of magnetic resonance enterography (MRE) were approached for potential recruitment. Thirteen potential participants were excluded because they had used antibiotics, probiotics, or prebiotics in the three months before inclusion ($n = 11$) or had other concomitant organic digestive diseases ($n = 2$), as determined from medical records and questionnaires administered during recruitment. Therefore, the final patient cohort contained 278 subjects (derivation cohort, $n = 214$; test cohort, $n = 64$). All patient analyses reported in this study were derived from the derivation cohort, while the test cohort was solely utilized to verify the generalization ability of our constructed models.

Simultaneously, 30 healthy controls (HCs) were recruited to provide fecal and serum samples based on open advertisements in the community. HCs were eligible to participate if they showed no gastrointestinal symptoms or antibiotic, probiotic, or prebiotic use in the three months before inclusion. The HCs were also clinically examined for diseases. Two outliers were excluded from HCs after 16S rRNA gene sequencing. Therefore, the final HC cohort comprised 28 subjects. In total, 306 subjects took part in this study. Their demographics, medical histories and other clinical variables were collected after enrollment according to standard procedures. Detailed clinical information about all study participants is provided in Table EV1.

## Collection of fecal and blood specimens

The participants provided fecal and/or serum samples at the hospital. Fecal samples were acquired in collection cups and immediately frozen at −80 °C until analysis. Blood samples were drawn after overnight fasting, sent directly to the laboratory for serum collection, and frozen at −80 °C until analysis. All 278 patients with CD provided blood samples; fecal samples were available for 216 of these patients (derivation cohort, $n = 152/214$; test cohort, $n = 64/64$). Both fecal and serum samples were analyzed for all 28 HCs.

## Magnetic resonance enterography

All enrolled patients with CD underwent MRE. Bowel preparation before the scanning procedure was performed as described previously (Li et al, 2017, 2018). MRE was performed using a 3.0 T magnetic resonance system (MAGNETOM Prisma; Siemens Healthineers, Erlangen, Germany) with multichannel phased-array body coils. The scan protocol (including T2-weighted imaging, magnetization transfer imaging (MTI), diffusion-weighted imaging, and pre-/post-enhancement T1-weighted imaging) was the same as that described in our previous study (Li et al, 2018). Among them, MTI was acquired using two gradient-echo datasets, with or without the application of an off-resonant prepulse (frequency offset: 1.2 kHz, duration: 9984 μs, effective flip angle: 500°, bandwidth: 192 Hz).

## Assessment and grading of intestinal fibrosis

The quantitative assessment of intestinal fibrosis was based on the normalized MTR derived from MTI as described previously (Li et al, 2018). In brief, the MTR was calculated using the following equation: MTR (%) = $[1 - (M_{sat}/M_0)] \times 100$, where $M_{sat}$ and $M_0$ are the signal intensities acquired with and without off-resonance prepulse saturation, respectively. In the MTR map, a region of interest was drawn at the midpoint level of the most severe bowel segment (i.e., the part of the intestine with the greatest stricture and/or the thickest bowel wall), including the full thickness of the whole circular bowel, to obtain the MTR of the bowel wall. The MTR of the skeletal muscle was also measured in the same MTR map, and the MTR of the bowel wall was divided by the MTR of the skeletal muscle to yield a normalized MTR, to minimize individual variation (Li et al, 2018). All image postprocessing and parameter calculations were conducted using Python (version 3.9).

A normalized MTR < 0.71 was defined as indicating none-mild bowel fibrosis (namely, BF1), and a normalized MTR ≥ 0.71 was defined as indicating moderate-severe bowel fibrosis (namely, BF2) (Li et al, 2018). According to the normalized MTR of the most severe bowel segment per patient, there were 93 patients with none-mild fibrosis and 121 patients with moderate-severe fibrosis in the derivation cohort, while in the test cohort, there were 20 patients with none-mild fibrosis and 44 patients with moderate-severe fibrosis. Given that CD is characterized by multiple segmental bowel lesions, we also stratified the patients according to the lesion with the highest normalized MTR value rather than the section with the most severe stricture/greatest thickness, to ensure that the fibrosis stratification method we chose was representative and reliable. Our results showed that the fibrosis classifications remained unchanged (Fig. EV1C,D; Appendix Table S4).

## Assessment of intestinal fibrosis-associated magnetic resonance enterography features

The occurrence and progression of intestinal fibrosis led to a series of changes in the intestine that can be observed by morphological and functional MRE (Bruining et al, 2018). To explore the relationship between the gut microbiota, metabolites, and intestinal fibrosis-associated imaging features, the MRE findings of the most severely diseased intestines (i.e., the part of the intestine with the greatest stricture and/or the thickest bowel wall) were jointly evaluated by a team of five professionals with experience in intestinal imaging using quantitative measurement or semiquantitative scoring methods. The interpreted imaging features included bowel stricture, penetrating disease, perienteric effusion, comb sign,

wall thickness, mural apparent diffusion coefficient, intramural edema, mural hyperenhancement, mural enhancement pattern, length of diseased bowel, adenopathy, and perianal diseases (Table EV10).

## 16S rRNA gene amplicon sequencing

All the samples were sequenced or measured in one batch, and processed using an identical bioinformatic pipeline.

Fecal samples collected from 216 patients with CD (derivation cohort, $n = 152$; test cohort, $n = 64$) and 28 HCs, as described in "Collection of fecal and blood specimens" above, were subjected to 16S rRNA gene amplicon sequencing. Total genomic DNA was extracted from the samples using the OMEGA Soil DNA Kit (M5635-02) (Omega Bio-Tek, Norcross, GA, USA) according to the manufacturer's instructions and was subsequently stored at −20 °C for further analysis. The quantity and quality of the extracted DNA were measured using a NanoDrop NC2000 spectrophotometer (Thermo Fisher Scientific, Waltham, MA, USA) and by agarose gel electrophoresis, respectively.

The V3-V4 regions of bacterial 16S rRNA genes were amplified by polymerase chain reaction (PCR) using the forward primer 338 F (5'-ACTCCTACGGGAGGCAGCA-3') and the reverse primer 806 R (5'-GGACTACHVGGGTWTCTAAT-3'). Sample-specific 7-bp barcodes were incorporated into the primers for multiplex sequencing. The PCR mixture contained 5 μl of buffer (5×), 0.25 μl of Fast pfu DNA Polymerase (5 U/μl), 2 μl (2.5 mM) of dNTPs, 1 μl (10 μM) of each forward and reverse primer, 2 μl of DNA template, and 8.75 μl of ddH$_2$O. The thermal cycling program consisted of initial denaturation at 98 °C for 2 min, followed by 25 cycles of denaturation at 98 °C for 15 s, annealing at 55 °C for 30 s, and extension at 72 °C for 30 s, with a final extension of 5 min at 72 °C. PCR amplicons were purified with Vazyme VAHTS DNA Clean Beads (Vazyme, Nanjing, China) and quantified using the Quant-iT PicoGreen dsDNA Assay Kit (Invitrogen, Carlsbad, CA, USA). After the individual quantification step, the amplicons were pooled in equal amounts, and pair-end 2 × 250 bp sequencing was performed on the Illumina NovaSeq platform with a NovaSeq 6000 SP Reagent Kit (500 cycles) at Shanghai Metabo-Profile Biotechnology Co., Ltd. (Shanghai, China).

Microbiome bioinformatics analysis was conducted using QIIME2 2019.4 (Bolyen et al, 2019) based on the official tutorials (https://docs.qiime2.org/2019.4/tutorials/). Briefly, raw sequence data were demultiplexed using the demux plugin, followed by primer cutting with the Cutadapt plugin (Kechin et al, 2017). Sequences were then quality-filtered, denoised, and merged, and chimeras were removed using the DADA2 plugin (Callahan et al, 2016). Non-singleton amplicon sequence variants (ASVs) were aligned with MAFFT (Katoh et al, 2002) and used to construct a phylogeny with FastTree2 (Price et al, 2009). Taxonomy was assigned to ASVs using the classify-sklearn naïve Bayes taxonomy classifier in the feature-classifier plugin (Bokulich et al, 2018) against the SILVA Release 132 Database (Koljalg et al, 2013).

## Metagenomic shotgun sequencing

Metagenomic shotgun sequencing was performed on fecal samples collected from 47 patients with CD (derivation cohort, $n = 36$; test cohort, $n = 11$) and 24 HCs to analyze the correlation between the microbial metabolization pathways and blood aspartic acid levels. These 71 subjects were randomly selected from the two cohorts in this study. Total microbial genomic DNA was extracted from the samples using the OMEGA Mag-Bind Soil DNA Kit (M5635-02) (Omega Bio-Tek, Norcross, GA, USA) following the manufacturer's instructions and stored at −20 °C prior to further assessment. The quantity and quality of the extracted DNA were measured using a Qubit™ 4 Fluorometer with Wi-Fi: Q33238 (Qubit™ Assay Tubes: Q32856; Qubit™ 1X dsDNA HS Assay Kit: Q33231) (Invitrogen, USA) and by agarose gel electrophoresis, respectively. The extracted microbial DNA was processed to construct metagenome shotgun sequencing libraries with an insert size of 400 bp with the Illumina TruSeq Nano DNA LT Library Preparation Kit. Each library was sequenced on the Illumina NovaSeq platform (Illumina, USA) according to the PE150 strategy at Metabo-Profile Biotechnology Co., Ltd. (Shanghai, China).

Raw sequencing reads were processed to obtain quality-filtered reads for further analysis. First, sequencing adapters were removed from the sequencing reads using Cutadapt (v1.2.1) (Martin, 2011). Second, low-quality reads were trimmed using a sliding-window algorithm in FASTP (Chen et al, 2018). Third, reads were aligned to the host genome using BMTagger to remove host contamination (Rotmistrovsky and Agarwala, 2011).

Once quality-filtered reads were obtained, the taxonomic classification of metagenomics sequencing reads from each sample was performed using Kraken2 (Wood et al, 2019) against a RefSeq-derived database that included genomes from archaea, bacteria, viruses, fungi, protozoans, metazoans and Viridiplantae. Reads assigned to metazoans or Viridiplantae were removed from the downstream analysis. Megahit (v1.1.2) (Li et al, 2015) was employed to assemble the results from each sample using the meta-large preset parameters. The generated contigs (longer than 300 bp) were then pooled together and clustered using MMseqs2 (Steinegger and Soding, 2017) in "easy-linclust" mode, setting the sequence identity threshold to 0.95 and coverage of the residues of the shorter contig to 90%.

The lowest common ancestor taxonomy of the nonredundant contigs was obtained by aligning them against the NCBI-nt database with MMseqs2 (Steinegger and Soding, 2017) in "taxonomy" mode, and contigs assigned to Viridiplantae or Metazoa were dropped in the following analysis. MetaGeneMark (Zhu et al, 2010) was used to predict genes in the contigs. The coding sequences (CDSs) of all samples were clustered by using MMseqs2 (Steinegger and Soding, 2017) in "easy-cluster" mode, setting the protein sequence identity threshold to 0.90 and the coverage of the residues of the shorter contig to 90%. To assess the abundances of these genes, the high-quality reads from each sample were mapped onto the predicted gene sequences using salmon (Patro et al, 2015) in quasi mapping-based mode with "--meta --minScoreFraction=0.55", and copy per kilobase per million mapped reads (CPM) method was used to normalize abundance values in metagenomes. The functionality of the nonredundant genes was evaluated by annotation using MMseqs2 (Steinegger and Soding, 2017) in "search" mode against the KEGG, EggNOG and CAZy protein databases. EggNOG and GO results were obtained using EggNOG-mapper (v2) (Cantalapiedra et al, 2021). GO results were evaluated using map2slim (www.metacpan.org). KO results were obtained using KOBAS (Bu et al, 2021).

## Fecal and blood metabolomics

As described in "Collection of fecal and blood specimens" above, a total of 216 patients with CD (derivation cohort, $n = 152$; test cohort, $n = 64$) and 28 HCs provided fecal samples for targeted metabolomics profiling, and blood samples for targeted metabolomics profiling were collected from all 306 participants.

The targeted metabolomics analysis of the samples was performed with Metabo-Profile (Shanghai, China). The sample preparation procedures were performed according to previously published methods with minor modifications (Xie et al, 2021). Briefly, for fecal-targeted metabolomics, 5 mg of feces was homogenized with zirconium oxide beads, while for blood-targeted metabolomics, 20 μL of serum was transferred to a 96-well plate. Subsequently, 120 μL of methanol containing an internal standard was used to extract the metabolites. The resulting supernatants were subjected to derivatization with 3-nitrophenylhydrazine (3-NPH) and N-(3-(dimethylamino)propyl)-N′-ethylcarbodiimide (EDC)·HCl (Sigma-Aldrich, St. Louis, MO, USA). Subsequently, the derivatized samples were analyzed by ultraperformance liquid chromatography coupled to tandem mass spectrometry (UPLC–MS/MS) (ACQUITY UPLC Xevo TQ-S, Waters Corp., Milford, MA, USA). All of the standards were obtained from Sigma-Aldrich (St. Louis, MO, USA), Steraloids Inc. (Newport, RI, USA), and TRC Chemicals (Toronto, ON, Canada). Quality control samples were prepared following the same procedures applied for the test samples and were injected at regular intervals to ensure instrument system stability. The raw data files generated by UPLC–MS/MS were processed using Targeted Metabolome Batch Quantification (TMBQ) software (v1.0, HMI, Shenzhen, Guangdong, China) to perform peak integration, calibration, and quantitation for each metabolite. The in-house-developed iMAP platform (v1.0, Metabo-Profile, Shanghai, China) was used for statistical analysis.

## Associations between L-aspartic acid, intestinal inflammation, and fibrosis

To investigate whether the relationship between L-aspartic acid and intestinal fibrosis was influenced by coexisting intestinal inflammation, we initially examined the variations in L-aspartic acid levels across different inflammatory groups. Considering that conventional endoscopy may fail to reach small intestinal lesions in some patients, a simplified magnetic resonance index of activity (MARIAs) (Ordas et al, 2019) derived from MRE was employed for the evaluation of intestinal inflammation. MARIAs = (1×thickness>3 mm)+(1×edema)+(1×fat stranding)+ (2×ulcers), where thickness refers to bowel wall thickness, edema indicates mural edema, and fat stranding means perienteric effusion in our study. The bowel site for intestinal inflammation assessment was chosen to be the same as that for fibrosis evaluation. A MARIAs≥2 was defined as moderate-severe bowel inflammation (namely, BI2), while a MARIAs<2 was defined as none-mild bowel inflammation (namely, BI1) (Ordas et al, 2019). Finally, 130 patients were categorized as having none-mild inflammation, and the other 84 patients exhibited moderate-severe inflammation in the derivation cohort. There was no significant difference in L-aspartic acid levels between BI1 and BI2 (Fig. EV3; Appendix Table S5). To elucidate the correlation between L-aspartic acid and both fibrosis and inflammation, a multiple linear regression model was constructed using the enter method. The model showed that L-aspartic acid was significantly associated with bowel fibrosis severity ($\beta = 2.088$, $P = 0.043$) but not with inflammation severity ($\beta = -0.513$, $P = 0.623$). The aforementioned findings collectively suggested that the association between L-aspartic acid and intestinal fibrosis remained unaltered in the presence of concurrent intestinal inflammation.

## In vivo rat colitis model experiments

### Animal model
Female Sprague–Dawley rats aged 6 weeks (Guangdong Medical Laboratory Animal Center, Guangzhou, China) were used to mimic intestinal fibrosis to different degrees. The animals were housed in individually ventilated cages, maintaining a temperature range of 22–26 °C with a maximum daily fluctuation of 4 °C. The relative humidity was maintained between 40 and 70%. A minimum of 15 times air changes per hour was ensured to maintain a barrier environment. The airflow velocity did not exceed 0.2 m/s. To establish this model (Dillman et al, 2013), 2,4,6-trinitrobenzene sulfonic acid (TNBS; Sigma-Aldrich, St. Louis, Missouri, USA) was administered at 150 mg/kg/week since L-aspartic acid was administered at 200 mg/kg every other day as referenced (Rao et al, 2021). The rats were allocated to three treatment groups: Tx (treated with TNBS and L-aspartic acid), control group 1 (C1, treated with TNBS), and control group 0 (C0, treated only with PBS). Specifically, the rats in the Tx group were treated with TNBS and L-aspartic acid for either 2 weeks ($n = 3$) or 3 weeks ($n = 6$), while those in the C1 group received TNBS treatment for either 2 weeks ($n = 6$) or 3 weeks ($n = 6$). Those in the C0 group were treated with PBS only for either 2 weeks ($n = 3$) or 3 weeks ($n = 3$).

### Micro-PET/CT and data analysis in animal experiments
The rats were administered a dose of 300 MBq of ${}^{18}$F-FAPI via tail vein injection (100–200 μL) one hour prior to scanning. PET-CT imaging was performed using a micro-PET system (Inviscan SAS, Strasbourg, France), and rats were then imaged for a 15-minute static acquisition (Li et al, 2012). Three-dimensional regions of interest (ROIs) were manually placed on the PET images in areas where the intestines exhibited the highest degree of fibrotic activity. The three-dimensional ROIs did not encompass the entire lesion volume to avoid partial volume effects. Subsequently, the maximum standardized uptake value (SUV$_{max}$) in the intestinal lesion was obtained for further analysis.

### MTI protocol and data analysis in animal experiments
The MTI scanning protocol was performed on the intestines of the rats according to a previously described method (Lu et al, 2021) and using a 3.0-T MR system (MAGNETOM Prisma; Siemens Healthineers, Erlangen, Germany) and a 4-channel animal coil (Shanghai Chenguang Medical Technology Co., Ltd., Shanghai, China). The postprocessing and data analysis methods employed for MTI in rats were the same as those previously described for patients with CD in this study.

### Pathological analysis
The rats were promptly euthanized following scanning in accordance with the treatment protocol for pathological

examination. Intestinal lesion segments from the Tx and C1 groups as well as normal colorectal segments from the C0 group were obtained and fixed using 4% paraformaldehyde. Following embedding in paraffin, the intestinal specimens were sectioned into 4-μm slices and subjected to staining with hematoxylin and eosin (HE) as well as Masson's trichrome (Li et al, 2018). HE staining was utilized to assess intestinal inflammation, while Masson's trichrome staining was employed for the evaluation of intestinal fibrosis using a semiquantitative scoring system (scores 0-4), as described in our previous study (Li et al, 2018).

## In vitro human primary intestinal myofibroblast experiments

### Isolation and culture of primary human intestinal myofibroblasts

Primary human intestinal myofibroblasts (HIMFs) were isolated from the surgically resected intestinal specimen of patients with CD, and cultured in Dulbecco's minimal essential medium supplemented with 10% fetal bovine serum (FBS, GIBCO, Life Technologies Corporation, Grand Island, NY, USA) and antibiotics, according to previously established protocols (Musso et al, 1999; Zhao et al, 2020). The cells were established as long-term cultures, which were fed twice weekly and sub-cultured at the confluence., and were utilized between passages 6 and 8.

### Cell treatment

To evaluate the profibrogenic properties of *L*-aspartic acid in vitro, primary HIMFs ($1.5 \times 10^5$ cells/well) were inoculated in a six-well plate and then treated with 5 ng/mL TGF-β1, either alone or cocultured with 0 μM, 6.5 μM or 13 μM *L*-aspartic acid in serum-free HIMF culture medium. After 48 h, HIMFs RNA and protein were examined by real-time RT-PCR and western blotting, respectively.

### qRT–PCR

Total RNA was extracted from the HIMFs using TRIzol reagent (Ambion, Carlsbad, CA). Then, equal amounts of RNA (1 μg) were reverse-transcribed into cDNA using the ReverTra Ace qPCR RT Master Mix Kit (TOYOBO, Osaka, Japan) according to the manufacturer's instructions. All qRT-PCR assays were performed using a Roche Light Cycler 96 instrument (Roche) with Faster Start Essential DNA Probes Master (Roche). The mRNA levels of all genes were normalized to that of β-actin. PCR amplification was performed with the following primers (5′ to 3′): Human collagen 1A1 (COL1A1) sense TAGTCTGTCCTGCGTCCTCT and antisense TTATGTTTGGGTCATTTCCA primers; Human fibronectin 1 (FN) sense CTACGGATGACTCGTGCTTT and antisense TTCCTTCTGCCACTGTTCTC primers; Human actin alpha 2 (ACTA2) sense CTGAGCGTGGCTATTCCTTC and antisense GCTGGAAGGTGGACAGAGAG primers; Human β-actin sense GTTGCTATCCAGGCTGTG and antisense GTTGCTATC-CAGGCTGTG primers.

### Western blotting

To obtain protein extracts for western blotting, cells were homogenized in lysis buffer (RIPA buffer containing protease and phosphatase inhibitors (Cell Signaling, Beverly, MA) followed by centrifugation to clear the lysate. Equal amounts of protein lysates were separated by SDS-PAGE, transferred to a PVDF membrane,

and incubated overnight with primary antibodies: COL1A1 (1:1000 dilution, #72026, Cell Signaling Technology), FN (1:1000 dilution, ab268020, Abcam, Cambridge, MA), and β-actin (1:1000 dilution, 66008-1-Ig, Proteintech). Proteins were visualized using the ECL detection system (Amersham Pharmacia Biotech, Piscataway, NJ) with the appropriate secondary antibodies: anti-rabbit HRP (1:2000 dilution, A11008, Invirtrogen) or anti-mouse HRP (1:2000 dilution, A32723, Invirtrogen).

## Statistical analysis

All analyses were performed in R (v.4.2.1). Microbial alpha diversity was calculated using the Simpson, Shannon, Chao1, and Pielou indices and taxonomic richness. The Wilcoxon test was used to compare alpha diversity differences. Bray−Curtis distances were employed to represent beta diversity, and the PERMANOVA test was performed with the *adonis* function in vegan (v.2.6) to assess the microbial variance explained by different fibrosis groups. Principal component analysis (PCA) was conducted for normalized metabolomics data. Statistical significance was determined according to an FDR < 0.1 (Benjamini−Hochberg procedure).

Four linear models were used to investigate BF2-specific microbial and metabolomic features. Each model was corrected for potential confounders, including age, sex, BMI, smoking, and disease location (only within patients with CD). All the patients did not use antibiotics within 3 months.

(1) Features ~ CD vs. HC + covariates
(2) Features ~ BF1 vs. HC + covariates
(3) Features ~ BF2 vs. HC + covariates
(4) Features ~ BF1 vs. BF2 + covariates.

Considering the limited sample size in model (4), we defined BF2-specific features as those results that passed the significance threshold in model (3) but were not present in models (1) and (2) (Guo et al, 2023).

To select BF2-associated morphological changes, we assessed twelve MRE features of intestinal lesions, including eight luminal (e.g., stricture, penetration, wall thickening) and four extraluminal (e.g., perienteric effusion, comb sign, adenopathy) features, based on the $\chi^2$ test. Stricture, penetration, effusion and the comb sign were further subjected to association tests with omics data.

To explore the disease-predictive potential of omics data, we incorporated six machine learning approaches, including the decision tree, random forest, gradient-boosted decision tree, extra tree, logistic regression and support vector machine method, included in the *caret* R package (v.6.0). These analyses were performed on a cohort of 152 patients with all the multiomics data. Five models were compared to classify CD patients versus HCs and BF2 patients versus BF1 patients among patients with CD, as follows:

(1) Groups ~ age + sex + BMI + smoking + disease location
(2) Groups ~ microbial taxa
(3) Groups ~ blood metabolites
(4) Groups ~ fecal metabolites
(5) Groups ~ all combined.

## The paper explained

### Problem

Intestinal fibrosis is the most important cause of disability and poor prognosis in patients with Crohn's disease. However, its profibrotic mechanisms remain incompletely understood, limiting the development of effective therapeutic strategies. The gut microbiota and its functional metabolites play crucial roles in the onset of Crohn's disease, suggesting their potential as targets for addressing intestinal fibrosis. While the associations of the microbiota and its metabolites in fibrogenesis have been studied in other organs, it has not yet been thoroughly explored in the gut.

### Results

In this study, a relatively comprehensive profile of specific microbes and metabolites associated with moderate-severe intestinal fibrosis has been revealed, offering potential targets for classifying this condition and conducting therapeutic trials to address it. A "bridge" between the small molecule profiles and the macroscopic imaging phenotypes of intestinal fibrosis is also established, emphasizing the potential to target these traits identified by multiomics analysis to effectively treat fibrosis. L-aspartic acid, identified as a potential target, has been demonstrated to induce fibrotic gene activation in HIMFs in vitro and promote the aggravation of intestinal fibrosis in rats in vivo.

### Impact

These findings presented here enhance our understanding of the potential pathogenic mechanism underlying intestinal fibrosis and offer promising prospects for advancing research in this field, potentially contributing to improving patient prognosis and reducing healthcare costs.

---

AUC comparisons were assessed using the DeLong test. To ensure model generalizability, the whole dataset was randomly split into training (70%) and validation (30%) parts. The model training was used fivefold cross-validation with parameters of train Control (method = "repeatedcv", repeats=5) in *caret* R package. Moreover, we validated the models in an independent external test cohort ($n = 64$).

To investigate the correlations between the four specific morphological changes on MRE images (i.e., stricture, penetration, perienteric effusion and the comb sign) and the omics data, we performed linear regression analysis adjusting for age, sex, BMI, smoking status and disease location in BF1 and BF2 patients. Only associations with a nominal significance of $P < 0.05$ were further included in a meta-analysis to evaluate their consistency between BF1 and BF2 patients. The results with meta $P$ values that passed FDR correction and showed a Q Test $P > 0.05$ were considered significant. The results with a Q test $P < 0.05$ were considered potentially heterogeneous between the two groups, indicating a fibrosis-dependent effect.

To identify putative links between the gut microbiota, metabolites and morphological changes, causal mediation analysis was performed using the *mediation* R package (v.4.2.1). We hypothesized that the blood and intestinal metabolites might be produced or metabolized by the gut microbiota, and the metabolites were therefore used as mediators. The gut microbiota was defined as exposures while stricture, penetration, perienteric effusion and the comb sign were defined as outcomes. $P$ values of average causal mediation effects (ACME) were used to assess

whether specific taxa potentially contributed to morphological features in a manner mediated specific small molecules. The *plotly* R package was used for visualization.

Both in vivo and in vitro results are presented as the mean ± the standard deviation (SD) and were analyzed with the Prism statistics software package version 8.0.0 (GraphPad, San Diego, CA). A *T* test was used to compare [18]F-FAPI uptake and normalized MTRs between groups of rats under different treatment strategies. The fibrotic scores of intestinal lesions in rats among different treatment groups were compared with Fisher's exact test. The densitometric data from the western blot assay and qRT-PCR data were analyzed via one-way analysis of variance (ANOVA) with Tukey's post hoc test. Differences were considered statistically significant at $P < 0.05$. No statistical methods were used to predetermine the sample size of participants. All animal experiments and in vitro studies were performed and analyzed in a blinded manner.

## For more information

Xuehua Li's lab website: https://www.researchgate.net/lab/Xuehua-Li-Lab.

## Data availability

The datasets and computer code produced in this study are available in the following databases: (1) The raw data of fecal microbiome sequencing: Sequence Read Archive (SRA) database; accession number: PRJNA1029396; https://www.ncbi.nlm.nih.gov/bioproject/1029396. (2) Bioinformatics analysis scripts: GitHub (https://github.com/WendyNice/IBD_fibrosis).

The source data of this paper are collected in the following database record: biostudies:S-SCDT-10_1038-S44321-024-00129-8.

## Peer review information

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

## Acknowledgements

This study was supported by National Key R&D Program of China (2023YFC2507300), Key-Area Research and Development Program of

Guangdong Province (2023B1111040003), National Natural Science Foundation of China (82270693, 82271958, 82070680, 82072002, 82222010, 82170537, 81970483), Guangdong Basic and Applied Basic Research Foundation (2023B1515020070, 2023A1515011097), and Fundamental Research Funds for the Central Universities, Sun Yat-sen University (24ykqb003). The funders of the study had no role in study design, data collection, data analysis, data interpretation, or writing of the report.

## Author contributions

**XueHua Li**: Conceptualization; Data curation; Formal analysis; Supervision; Funding acquisition; Validation; Investigation; Methodology; Writing—original draft; Project administration; Writing—review and editing. **Shixian Hu**: Conceptualization; Data curation; Software; Formal analysis; Validation; Investigation; Methodology; Writing—original draft; Writing—review and editing. **Xiaodi Shen**: Conceptualization; Data curation; Formal analysis; Validation; Investigation; Methodology; Writing—original draft; Writing—review and editing. **Ruonan Zhang**: Conceptualization; Data curation; Formal analysis; Validation; Investigation; Methodology; Writing—original draft; Writing—review and editing. **Caiguang Liu**: Data curation; Formal analysis; Validation; Investigation; Methodology; Writing—original draft; Writing—review and editing. **Lin Xiao**: Data curation; Formal analysis; Validation; Investigation; Methodology; Writing—original draft; Writing—review and editing. **Jinjiang Lin**: Data curation; Validation; Investigation; Writing—original draft; Writing—review and editing. **Li Huang**: Data curation; Validation; Investigation; Writing—original draft; Writing—review and editing. **Weitao He**: Data curation; Validation; Investigation; Writing—original draft; Writing—review and editing. **Xinyue Wang**: Data curation; Validation; Investigation; Writing—original draft. **Lili Huang**: Data curation; Validation; Investigation; Writing—original draft. **Qingzhu Zheng**: Data curation; Validation; Investigation; Writing—original draft. **Luyao Wu**: Data curation; Validation; Investigation; Writing—original draft. **Canhui Sun**: Methodology; Writing—original draft; Project administration. **Zhenpeng Peng**: Methodology; Project administration; Writing—review and editing. **Minhu Chen**: Methodology; Project administration; Writing—review and editing. **Ziping Li**: Methodology; Project administration; Writing—review and editing. **Rui Feng**: Methodology; Writing—review and editing. **Yijun Zhu**: Methodology; Writing—review and editing. **Yangdi Wang**: Conceptualization; Data curation; Formal analysis; Validation; Investigation; Methodology; Writing—original draft; Writing—review and editing. **Zhoulei Li**: Conceptualization; Data curation; Formal analysis; Validation; Investigation; Methodology; Writing—original draft; Writing—review and editing. **Ren Mao**: Conceptualization; Data curation; Formal analysis; Supervision; Funding acquisition; Validation; Investigation; Methodology; Writing—original draft; Project administration; Writing—review and editing. **Shi-Ting Feng**: Conceptualization; Data curation; Formal analysis; Supervision; Funding acquisition; Validation; Investigation; Methodology; Writing—original draft; Project administration; Writing—review and editing.

Source data underlying figure panels in this paper may have individual authorship assigned. Where available, figure panel/source data authorship is listed in the following database record: biostudies:S-SCDT-10_1038-S44321-024-00129-8.

## Disclosure and competing interests statement

The authors declare no competing interests.

# Expanded View Figures

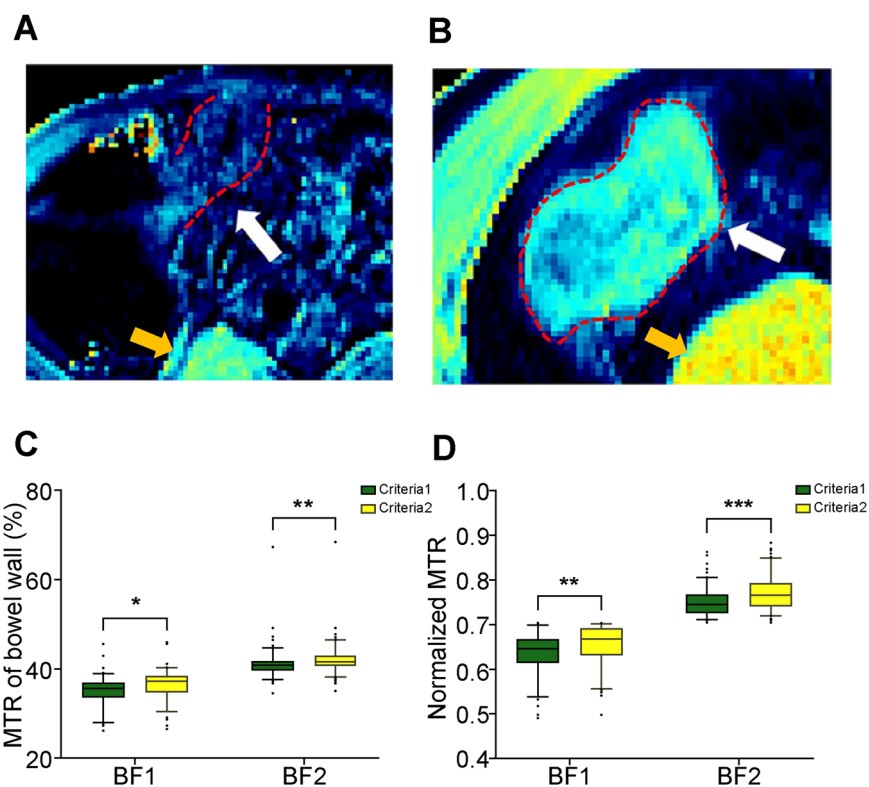

**Figure EV1. Assessment of intestinal fibrosis using MTI.**

(A) Axial color MTR map demonstrating a significant reduction in the MT effect in the terminal ileum in CD (white arrow and red dashed line) compared to that in the right psoas muscle (yellow arrow), indicating none-mild fibrosis (normalized MTR = 0.63). (B) Another example demonstrating an MT effect in the terminal ileum in CD (white arrow and red dashed line) that is comparable to that in the right psoas muscle (yellow arrow), indicating moderate-severe fibrosis (normalized MTR = 0.81). (C, D) Although there are significant differences in the bowel MTR (C) and the normalized MTR (D) between criterion 1 (normalized MTR from the most severe lesion) and criterion 2 (highest normalized MTR from all lesions) by $T$ test in either BF1 ($n = 93$) or BF2 ($n = 121$) patients (all $P < 0.05$), the fibrosis classification of each patient remains consistent regardless of which normalized MTR selection criterion is used. Boxplots (C, D) represent the interquartile ranges (25th through 75th percentiles, boxes), medians (50th percentiles, bars within the boxes), and the 5th and 95th percentiles (whiskers below and above the boxes). *$P < 0.05$; **$P < 0.01$; ***$P < 0.001$. The dots in plots indicate the outliers. Source data are available online for this figure.

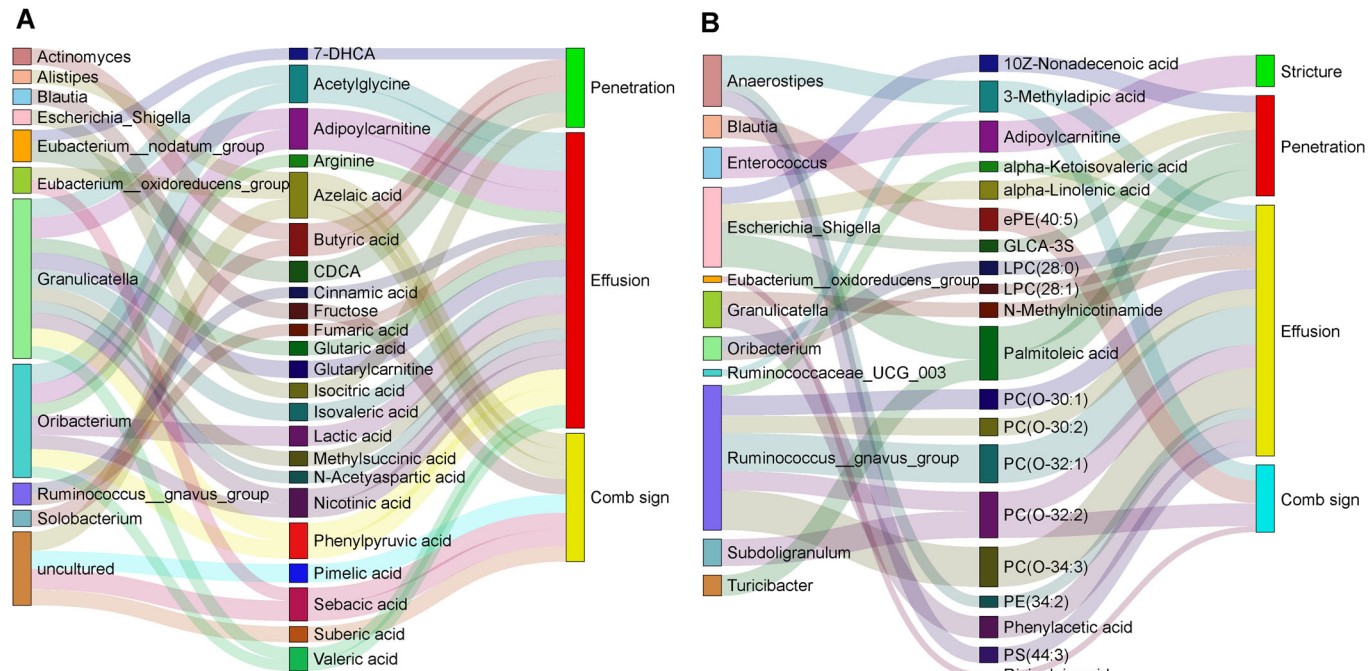

**Figure EV2. Putative links between the gut microbiota, metabolites and MRE features in BF1 patients.**

(A, B) Sankey plots indicating bacterial contributors to MRE features mediated by fecal (n = 23; **A**) and blood metabolites (n = 20; **B**) by mediation analysis. All P values of average causal mediation effects <0.05. Color boxes represent the gut microbiota (left column), metabolites (intermediate column) and MRE features (right column), respectively. The line width represents the relative size of P value for the correlation between the two "boxes" (wider lines indicate relatively smaller P -values). Source data are available online for this figure.

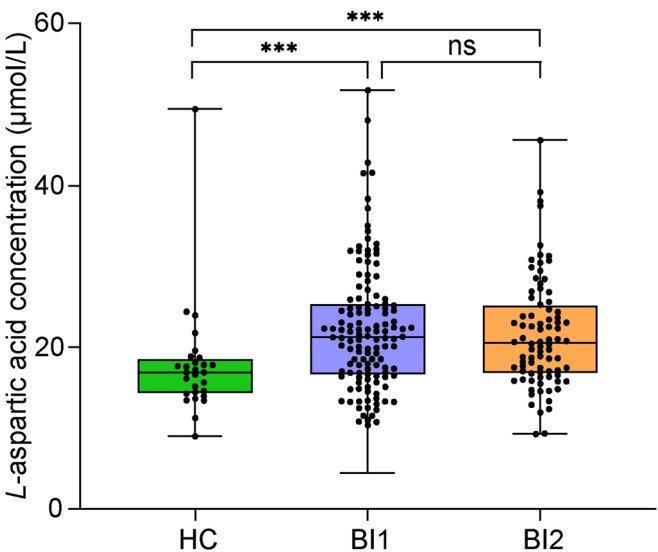

**Figure EV3. Comparison of *L*-aspartic acid levels in blood among HCs, CD patients with none-mild inflammation, and CD patients with moderate-severe inflammation.**

A total of 130 patients with CD are categorized as having none-mild inflammation (BI1), and the other 84 patients exhibit moderate-sever inflammation (BI2). The boxplots show significant differences in *L*-aspartic acid levels between HCs and BI1 patients ([17.78 ± 7.07] μmol/L vs. [22.12 ± 7.90] μmol/L, *P* < 0.001) as well as between HCs and BI2 patients ([17.78 ± 7.07] μmol/L vs. [21.59 ± 6.80] μmol/L, *P* < 0.001) analyzed by the Mann–Whitney *U* test. However, no significant difference in *L*-aspartic acid levels is observed between BI1 and BI2 patients ([22.12 ± 7.90] μmol/L vs. [21.59 ± 6.80] μmol/L, *P* = 0.708), suggesting that the severity of intestinal inflammation has no significant impact on the *L*-aspartate levels observed in this patient cohort. Boxplots represent the interquartile ranges (25th through 75th percentiles, boxes), medians (50th percentiles, bars within the boxes), and the 5th and 95th percentiles (whiskers below and above the boxes). ****P* < 0.001; ns, not significant. The dots in the plots indicate individual participants. Source data are available online for this figure.

