## [Peer Review File · EMBO Molecular Medicine]

Multimomics reveals microbial metabolites as key actors in intestinal fibrosis in Crohn's disease

XueHua Li, Shixian Hu, Xiaodi Shen, Ruonan Zhang, Caiguang Liu, Lin Xiao, Jinjiang Lin, Li Huang, Weitao He, Xinyue Wang, Lili Huang, Qingzhu Zheng, Luyao Wu, Canhui Sun, Zhenpeng Peng, Minhu Chen, Ziping Li, Rui Feng, Yijun Zhu, Yangdi Wang, Zhoulei Li, Ren Mao, and Shi-Ting Feng

Corresponding authors: Shi-Ting Feng (fengsht@mail.sysu.edu.cn) , Yangdi Wang (wangyd83@mail.sysu.edu.cn), Zhoulei Li (lizhlei3@mail.sysu.edu.cn), Ren Mao (maor5@mail.sysu.edu.cn)

Review Timeline:

Submission Date:	25th Feb 24
Editorial Decision:	30th Apr 24
Revision Received:	27th Jun 24
Editorial Decision:	17th Jul 24
Revision Received:	2nd Aug 24
Accepted:	13th Aug 24

Editor: Zeljko Durdevic

Transaction Report:

30th Apr 2024

Dear Prof. Li,

Thank you for the submission of your manuscript to EMBO Molecular Medicine, and please accept my apologies for the unusual delay in getting back to you. As I explained earlier one of the referees is not responding to our reminders to submit his/her report. Therefore, we had to invite additional reviewers, to have at least 2 reports to be able to make informed decision on your manuscript.

We have now received feedback from two of the three reviewers who agreed to evaluate your manuscript. As you will see from their reports pasted below, while both referees support publication of the manuscript, referee #3 also raises important concerns that should be addressed in a major revision of the current manuscript. If you would like to discuss further the points raised by the referees, I am available to do so via email or video. Let me know if you are interested in this option.

We would welcome the submission of a revised version within three months for further consideration. Please let us know if you require longer to complete the revision.

I look forward to receiving your revised manuscript.

Yours sincerely,

Zeljko Durdevic

We require:

2) Individual production quality figure files as .eps, .tif, .jpg (one file per figure). For guidance, download the 'Figure Guide PDF': (<https://www.embopress.org/page/journal/17574684/authorguide#figureformat>).

3) A .docx formatted letter INCLUDING the reviewers' reports and your detailed point-by-point responses to their comments. As part of the EMBO Press transparent editorial process, the point-by-point response is part of the Review Process File (RPF), which will be published alongside your paper.

4) A complete author checklist, which you can download from our author guidelines (<https://www.embopress.org/page/journal/17574684/authorguide#submissionofrevisions>). Please insert information in the checklist that is also reflected in the manuscript. The completed author checklist will also be part of the RPF.

6) It is mandatory to include a 'Data Availability' section after the Materials and Methods. Before submitting your revision, primary datasets produced in this study need to be deposited in an appropriate public database, and the accession numbers and database listed under 'Data Availability'. Please remember to provide a reviewer password if the datasets are not yet public (see <https://www.embopress.org/page/journal/17574684/authorguide#dataavailability>).

13) Author contributions: You will be asked to provide CRediT (Contributor Role Taxonomy) terms in the submission system.

These replace a narrative author contribution section in the manuscript.

14) A Conflict of Interest statement should be provided in the main text.

15) Every published paper now includes a 'Synopsis' to further enhance discoverability. Synopses are displayed on the journal webpage and are freely accessible to all readers. They include a short stand first (maximum of 300 characters, including space) as well as 2-5 one-sentence bullet points that summarize the paper. Please write the bullet points to summarize the key NEW findings. They should be designed to be complementary to the abstract - i.e. not repeat the same text. We encourage inclusion of key acronyms and quantitative information (maximum of 30 words / bullet point). Please use the passive voice. Please attach these in a separate file or send them by email, we will incorporate them accordingly.

Please also suggest a striking image or visual abstract to illustrate your article as a PNG file 550 px wide x 300-800 px high.

***** Reviewer's comments *****

Referee #2 (Comments on Novelty/Model System for Author):

This paper presents a proof-of-concept study which is the culmination of a program investigating the assessment of intestinal fibrosis radiologically that enables an understanding the potential causes or factors that promote fibrosis using metabolomics and studies of the microbiota. The techniques are sophisticated and the findings are convincing.

Referee #2 (Remarks for Author):

This is a significant and evolving body of work using sophisticated techniques to provide convincing proof-of-principle that the metabolic products of intestinal bacteria promote fibrosis in CD patients. It has identified new therapeutic strategies for the treatment/prevention of fibrosis. The next steps are to show that the biomarkers and proposed treatments have clinical utility in prospective studies [which is not always the case as the recently published PROFILE study of "top-down" infliximab demonstrates]. Nevertheless, fibrosis in CD is a major cause of morbidity and this study has made very important advances in our understanding of the pathogenic processes.

Referee #3 (Remarks for Author):

The article provides a detailed analysis of 278 CD patients and 28 healthy control subjects, with a large sample size that enhances the reliability of the study. Advanced multi-omics techniques and magnetic resonance enterography (MRE) were used to assess intestinal fibrosis, and the scientific validity and innovation of these methods are commendable. The study revealed associations between specific gut microbiota and metabolites with intestinal fibrosis, particularly the role of L-aspartic acid, providing a new direction for biomarker research in CD. They selected L-aspartic acid as an example among many targets, and obtained evidence that L-aspartic acid induced fibrosis gene activation in HIMFs in vitro and promoted intestinal fibrosis in rats in vivo, but the specific mechanism behind this effect was not explored in depth. In addition, there are still some issues that need to be resolved.

1. Has the author considered the potential impact of variability factors in sample collection, storage, and processing on the results?
2. Were appropriate statistical methods used to correct for multiple testing during multi-omics data analysis?
3. In the study of the causal relationship of L-aspartic acid, did the author consider other potential confounding factors such as patient diet, lifestyle, or medication?
4. Was the machine learning model mentioned in the article evaluated for its generalization ability through proper cross-validation?
5. For experiments involving L-aspartic acid in vitro and in vivo, it is suggested that the authors consider conducting more dose-response experiments to determine the specific mechanisms by which it affects intestinal fibrosis.
 - a) How were the concentrations of L-aspartic acid (0, 6.5, or 13 μ M) determined in HIMFs?
 - b) The study provides evidence of L-aspartic acid promoting intestinal fibrosis, but further experiments may be needed to elucidate its specific mechanisms of action.
 - c) While the study investigated the promoting effect of L-aspartic acid on intestinal fibrosis, it did not mention the use of potential inhibitors or other interventions to reverse or slow down the fibrotic process. Testing known anti-fibrotic drugs or new

compounds to see if they can counteract the effects of L-aspartic acid could be beneficial.

6. Does the article discuss the limitations of the study, such as sample selection bias, potential confounding factors, and the generalizability of the results?

7. The overall structure of the article is clear, but certain sections may need further refinement to enhance readability.

a) In Figure 1d, the authors describe that Xylose exhibited significant abundance in BF2 patients, but without discussing the role of Xylose in intestinal fibrosis in the manuscript. Additionally, it is difficult to discern from Figure 1d alone that Xylose was significantly more abundant in BF2 patients compared to the other two groups. Similarly, Figure 1e does not make it clear that L-aspartic acid levels in blood samples specifically decreased significantly in the BF2 group. To improve clarity, the authors could have included p-values alongside the FDR in Figures 1c-e to better illustrate the statistical significance between the three groups.

b) The stool and blood samples should be clearly marked in Figure 4d and 4e, respectively. The authors can also describe the results of Figure 4e in the manuscript and explain the relationship between the predicted microbial pathways and fecal aspartate levels.

Manuscript EMM-2024-19537

We thank the editor and reviewers for taking the time to assess our manuscript and for their constructive comments. We have updated the data, performed additional analyses and revised the manuscript accordingly. We believe our study is unique with first integration of gut microbiota, fecal and blood metabolites with MRE-detectable intestinal morphology to investigate the mechanisms underlying intestinal fibrosis in Crohn's disease. Please find below our point-by-point responses.

Comments from the Editor

When submitting your revised manuscript, please carefully review the instructions that follow below.

We perform an initial quality control of all revised manuscripts before re-review; failure to include requested items will delay the evaluation of your revision.

We require:

Comment 1. A .docx formatted version of the manuscript text (including legends for main figures, EV figures and tables). Please make sure that the changes are highlighted to be clearly visible.

Response: We appreciate the editor again for dedicating the time and effort in reviewing our manuscript. The changes in the manuscript have been highlighted using visible traces of modification.

Comment 2. Individual production quality figure files as .eps, .tif, .jpg (one file per figure). For

guidance, download the 'Figure Guide PDF':

(<https://www.embopress.org/page/journal/17574684/authorguide#figureformat>).

Response: We have prepared the individual production quality figure files according to the journal's guidance.

Comment 3. A .docx formatted letter INCLUDING the reviewers' reports and your detailed point-by-point responses to their comments. As part of the EMBO Press transparent editorial process, the point-by-point response is part of the Review Process File (RPF), which will be published alongside your paper.

Response: According to the editor's suggestion, we have prepared a response letter that includes the comments from the editor and reviewers, as well as our point-by-point responses to these comments.

Comment 4. A complete author checklist, which you can download from our author guidelines (<https://www.embopress.org/page/journal/17574684/authorguide#submissionofrevisions>). Please insert information in the checklist that is also reflected in the manuscript. The completed author checklist will also be part of the RPF.

Response: We have inserted the corresponding manuscript information in the author checklist accordingly.

Comment 5. Please note that all corresponding authors are required to supply an ORCID ID for their name upon submission of a revised manuscript.

Response: The ORCID ID of all the corresponding authors has been added in the Title Page.

Comment 6. It is mandatory to include a 'Data Availability' section after the Materials and Methods. Before submitting your revision, primary datasets produced in this study need to be deposited in an appropriate public database, and the accession numbers and database listed under 'Data Availability'. Please remember to provide a reviewer password if the datasets are not yet public (see <https://www.embopress.org/page/journal/17574684/authorguide#dataavailability>).

Response: The primary datasets used in this study have been deposited in the Sequence Read Archive (SRA) database, alongside the bioinformatics analysis scripts which have been stored on GitHub. Comprehensive information regarding these resources has been incorporated into the Data Availability Section.

Comment 7. For data quantification: please specify the name of the statistical test used to generate error bars and P values, the number (n) of independent experiments (specify technical or biological replicates) underlying each data point and the test used to calculate p-values in each figure legend. The figure legends should contain a basic description of n, P and the test applied. Graphs must include a description of the bars and the error bars (s.d., s.e.m.). See also 'Figure Legend' guidelines: <https://www.embopress.org/page/journal/17574684/authorguide#figureformat>

Response: According to the editor's suggestion and the guidelines for Figure Legend, we have thoroughly reviewed all figure legends and added any missing information as necessary.

Comment 8. At EMBO Press we ask authors to provide source data for the main manuscript figures. Our source data coordinator will contact you to discuss which figure panels we would need source data for and will also provide you with helpful tips on how to upload and organize the files.

Response: According to the source data coordinator's suggestion, we have prepared the source data for the main manuscript figures and included them in the revision materials.

Comment 9. Our journal encourages inclusion of *data citations in the reference list* to directly cite datasets that were re-used and obtained from public databases. Data citations in the article text are distinct from normal bibliographical citations and should directly link to the database records from which the data can be accessed. In the main text, data citations are formatted as follows: "Data ref: Smith et al, 2001" or "Data ref: NCBI Sequence Read Archive PRJNA342805, 2017". In the Reference list, data citations must be labeled with "[DATASET]". A data reference must provide the database name, accession number/identifiers and a resolvable link to the landing page from which the data can be accessed at the end of the reference. Further instructions are available at <https://www.embopress.org/page/journal/17574684/authorguide#referencesformat>.

Response: We appreciate the editor's suggestion. In this study, we did not use data sourced from public databases; hence, we did not include any information regarding citations from such databases.

Comment 10. We replaced Supplementary Information with Expanded View (EV) Figures and Tables that are collapsible/expandable online. A maximum of 5 EV Figures can be typeset. EV Figures should be cited as 'Figure EV1, Figure EV2' etc... in the text and their respective legends

should be included in the main text after the legends of regular figures.

- Additional Tables/Datasets should be labeled and referred to as Table EV1, Dataset EV1, etc.

Legends have to be provided in a separate tab in case of .xls files. Alternatively, the legend can be supplied as a separate text file (README) and zipped together with the Table/Dataset file. See

detailed instructions here:

<https://www.embopress.org/page/journal/17574684/authorguide#expandedview>.

Response: The original “Extended Data Figs. 1-3” have been revised to “Figs. EV 1-3” in the manuscript, and their respective legends have been included in the main text following the legends of regular figures. The original “Supplementary Tables 1-22” have also been revised to “Tables EV 1-22”. Their legends are provided in a separate tab in case of the Excel files.

Additionally, we provided a classifier for distinguishing the severity of intestinal fibrosis in patients with CD, originally labelled as "Supplementary data" but now renamed as "Classifier EV1". Its legend is presented as a separate text file (README) and zipped together along with this Classifier.

Comment 11. The paper explained: EMBO Molecular Medicine articles are accompanied by a summary of the articles to emphasize the major findings in the paper and their medical implications for the non-specialist reader. Please provide a draft summary of your article highlighting

This may be edited to ensure that readers understand the significance and context of the research.

Please refer to any of our published articles for an example.

Response: According to the editor's suggestion, we have added "The Paper Explained" to emphasize the major findings in our study and their medical implications.

Comment 12. For more information: There is space at the end of each article to list relevant web links for further consultation by our readers. Could you identify some relevant ones and provide such information as well? Some examples are patient associations, relevant databases, OMIM/proteins/genes links, author's websites, etc...

Response: We have added a website to introduce the corresponding author's lab, which specializes in advanced imaging research on inflammatory bowel disease.

Comment 13. Author contributions: You will be asked to provide CRediT (Contributor Role Taxonomy) terms in the submission system. These replace a narrative author contribution section in the manuscript.

Response: We have provided CRediT terms for each author in the submission system and updated this information in "Author Contributions" section of the main text.

Comment 14. A Conflict of Interest statement should be provided in the main text.

Response: A COI statement had been provided in the "Disclosure and competing interest statement" section in the main text.

Comment 15. Every published paper now includes a 'Synopsis' to further enhance discoverability. Synopses are displayed on the journal webpage and are freely accessible to all readers. They include

a short stand first (maximum of 300 characters, including space) as well as 2-5 one-sentences bullet points that summarizes the paper. Please write the bullet points to summarize the key NEW findings. They should be designed to be complementary to the abstract - i.e. not repeat the same text. We encourage inclusion of key acronyms and quantitative information (maximum of 30 words / bullet point). Please use the passive voice. Please attach these in a separate file or send them by email, we will incorporate them accordingly.

Please also suggest a striking image or visual abstract to illustrate your article as a PNG file 550 px wide x 300-800 px high.

Response: The "Synopsis" section, along with a graphic abstract, has already been prepared as per the editor's suggestion and has been included in the revision materials.

Comments from the reviewers

Reviewer 2

Comment 1 (Comments on Novelty/Model System for Author):

This paper presents a proof-of-concept study which is the culmination of a program investigating the assessment of intestinal fibrosis radiologically that enables an understanding the potential causes or factors that promote fibrosis using metabolomics and studies of the microbiota. The techniques are sophisticated and the findings are convincing.

Response: Thank you sincerely for your time and effort in reviewing our manuscript. We greatly appreciate your support and agreement with the conclusions drawn from our study.

Comment 2 (Remarks for Author):

This is a significant and evolving body of work using sophisticated techniques to provide convincing proof-of-principle that the metabolic products of intestinal bacteria promote fibrosis in CD patients. It has identified new therapeutic strategies for the treatment/prevention of fibrosis. The next steps are to show that the biomarkers and proposed treatments have clinical utility in prospective studies [which is not always the case as the recently published PROFILE study of "top-down" infliximab demonstrates]. Nevertheless, fibrosis in CD is a major cause of morbidity and this study has made very important advances in our understanding of the pathogenic processes.

Response: We appreciate the reviewer's comment. In our study, a relatively comprehensive profile of specific microbes and metabolites associated with moderate-severe intestinal fibrosis had been revealed, offering potential treatment targets for this condition. To validate the efficacy of these targets, we had selected *L*-aspartic acid as a representative factor to investigate its role in promoting fibrosis through both *in vivo* and *in vitro* studies. We totally agree with the reviewer that further investigation is necessary to determine the clinical utility of these biomarkers through prospective studies. We had included this information as future prospects in the last paragraph in the Discussion section.

Reviewer 3

(Remarks for Author):

The article provides a detailed analysis of 278 CD patients and 28 healthy control subjects, with a large sample size that enhances the reliability of the study. Advanced multi-omics techniques and magnetic resonance enterography (MRE) were used to assess intestinal fibrosis, and the scientific validity and innovation of these methods are commendable. The study revealed associations between

specific gut microbiota and metabolites with intestinal fibrosis, particularly the role of *L*-aspartic acid, providing a new direction for biomarker research in CD. They selected *L*-aspartic acid as an example among many targets, and obtained evidence that *L*-aspartic acid induced fibrosis gene activation in HIMFs in vitro and promoted intestinal fibrosis in rats in vivo, but the specific mechanism behind this effect was not explored in depth. In addition, there are still some issues that need to be resolved.

Response: Thank you for your time and effort in reviewing our manuscript to help improve it.

Comment 1. Has the author considered the potential impact of variability factors in sample collection, storage, and processing on the results?

Response: We appreciate the reviewer's comment. The controlling of potential factors during sample collection, storage and processing has been published in our previous study "Braun, T., Feng, R., Amir, A. *et al.* Diet-omics in the Study of Urban and Rural Crohn disease Evolution (SOURCE) cohort. *Nat Commun* **15**, 3764 (2024)." Fecal samples were acquired in collection cups and immediately frozen at -80°C . Blood samples were drawn after overnight fasting, sent directly to the laboratory for serum collection, and frozen at -80°C . All the samples were sequenced or measured in one batch, and processed using an identical bioinformatic pipeline. This information had been detailed and is now further supplemented in the Methods and Materials section.

Comment 2. Were appropriate statistical methods used to correct for multiple testing during multi-omics data analysis?

Response: Thank you for this question. To control false positives, we applied FDR correction during omics data analysis (Benjamini Hochberg approach) for each hypothesis.

- 1) To test microbial and metabolic features associated with intestinal fibrosis, we used four different models and considered all the test numbers for taxa and metabolites separately, with an FDR <0.1 as significance.
- 2) To test omics predictive potentials, we applied six machine learning models and used $P < 0.05$ for DeLong test comparing AUCs.
- 3) To test omics features associated with morphological alterations, we applied linear models in BF1 and BF2 groups and used FDR <0.1 as significant threshold.
- 4) To test the triangular relationships between microbiota, metabolites and morphological features, we applied causal mediation models with FDR <0.1 as significance considering all the test numbers.

The above information had been thoroughly described in the Statistical analysis section of the Methods.

Comment 3. In the study of the causal relationship of *L*-aspartic acid, did the author consider other potential confounding factors such as patient diet, lifestyle, or medication?

Response: We appreciate the reviewer's comment. In this study of the causal relationship of *L*-aspartic acid, we had taken these confounding factors in consideration. The subsequent explanations for the three confounding factors proposed by the reviewer are presented below:

- 1) Diet: we acknowledge that diet has significant influence on gut microbiota and CD risk. However, *L*-aspartic acid is commonly found in various food categories, including meat and plants¹. Therefore, accurately assessing the impact of diet on *L*-aspartic acid intake can be challenging. Moreover, there is limited research available on the association between diet and intestinal

fibrosis². To assess the causality between *L*-aspartic acid and intestinal fibrosis, we had provided both detail statistical and experimental evidence. Nevertheless, we also fully agree with the reviewer that inclusion of patients' dietary information can enhance our study conclusion. We have included it as a study limitation in the Discussion section.

2) Lifestyle: lifestyle has been reported to influence gut microbiota and thus, may play a role in shaping the metabolites³. The participants included in our study (both discovery and replication cohorts) were recruited from a highly reputable Chinese IBD center with wide geographical representation. Thus, the participants exhibit diverse lifestyles. Accurately assessing the impact of lifestyle on *L*-aspartic acid intake can also be challenging. However, we acknowledge the importance of considering this factor in future research and have addressed it as another study limitation in the Discussion section.

3) Medication: all the participants in our study did not use any antibiotics, probiotics, or prebiotics in the three months before inclusion. This information had been presented in “Human participants” subsection of the Methods section.

1. Górska-Warsewicz, H., Laskowski, W., Kulykovets, O., Kudlińska-Chylak, A., Czczotko, M., & Rejman, K. (2018). Food products as sources of protein and amino acids—The case of Poland. *Nutrients*, *10*(12), 1977.

2. Marion-Letellier, R., Leboutte, M., Amamou, A., Raman, M., Savoye, G., & Ghosh, S. (2021). Diet in Intestinal Fibrosis: A Double-Edged Sword. *Nutrients*, *13*(9), 3148.

3. Gacesa, R., Kurilshikov, A., Vich Vila, A., Sinha, T., Klaassen, M. A., Bolte, L. A., ... & Weersma, R. K. (2022). Environmental factors shaping the gut microbiome in a Dutch population. *Nature*, *604*(7907), 732-739.

Comment 4. Was the machine learning model mentioned in the article evaluated for its generalization ability through proper cross-validation?

Response: We apologize for the unclear description. We had used five-fold cross-validation for each

machine learning model training. More detail information has been added in the Statistical analysis subsection of the Methods section.

Comment 5. For experiments involving *L*-aspartic acid *in vitro* and *in vivo*, it is suggested that the authors consider conducting more dose-response experiments to determine the specific mechanisms by which it affects intestinal fibrosis.

a) How were the concentrations of *L*-aspartic acid (0, 6.5, or 13 μM) determined in HIMFs?

Response: Thank you for this constructive suggestion. To determine the appropriate concentrations of *L*-aspartic acid, we conducted a comprehensive review of the current literature on the utilization of *L*-aspartic acid and its isomer aspartate in *in vitro* studies (shown in Table R). As per the suggestion, we have added another two concentrations (3 μM , 30 μM) alongside 0 μM , 6.5 μM and 13 μM of *L*-aspartic acid for *in vitro* studies. As shown in Figure R, *L*-aspartic acid is found to promote TGF- β 1-induced fibrogenic activation in a trend of dose-dependent manner. However, there is no significant difference between the groups of cells treated with 0 μM and 3 μM *L*-aspartic acid. Similarly, compared to the treatment with 13 μM *L*-aspartic acid, the use of 30 μM *L*-aspartic acid does not enhance the pro-fibrotic effect induced by TGF- β 1. Hence, we consider that the use of 0 μM , 6.5 μM or 13 μM *L*-aspartic acid might be appropriate for investigating its pro-fibrotic role in HIMFs, which is consistent with previous studies (shown in Table R). The data has been updated in Figure 6.

Table R. The concentration gradient of *L*-aspartic acid in cell experiments.

Drugs	Cells	Concentration	Effects	References
-------	-------	---------------	---------	------------

Poly-L-aspartic Acid	HEK293T cells	0, 6.5, 13uM	Enhancing the suppression in a cystic fibrosis mouse model	J Biol Chem 2009 Mar13;284(11): 6885-92
L-aspartic acid	RPE cell	0, 0.0625, 0.125, 0.25 mM	Attenuating choroidal neovascularization via anti-inflammation	Exp Eye Res 2019 May: 182:93-100
Aspartate	Colonic organoids	0, 0.01, 0.1, 0.5, 1, 5 mM	Promoting ISCs proliferation and differentiation	Mol Nutr Food Res 2022 Dec;66(24): e2200168
Aspartate	Cancer cells lines	0, 150 μM	Limiting the cancer cell proliferation	Nat Cell Biol 2018 Jul;20(7):775-781
Aspartate	Cancer cells lines	0, 1, 10, 100, 1000uM	Limiting the tumour growth	Cell Biol 2018. Jul;20(7):782-788

Figure R. The use of different concentrations of *L*-aspartic acid promote the progression of intestinal fibrosis *in vitro*.

L-aspartic acid promoted TGF-β1-induced fibrogenic activation of human intestinal myofibroblasts (HIMFs). HIMFs were treated with TGF-β1 (5 ng/mL) and cocultured with different concentration of *L*-aspartic acid (0 μM, 3 μM, 6.5 μM, 13 μM or 30 μM). a-c, Results of qPCR analysis of the relative mRNA expression of collagen 1A1 (*COL1A1*), fibronectin (*FNI*), and actin alpha 2 (*ACTA2*). d, Representative Western blots showing the protein expression of COL1A1 and FN with β-actin as a loading control. Data are expressed as the mean ± SD.

b) The study provides evidence of *L*-aspartic acid promoting intestinal fibrosis, but further experiments may be needed to elucidate its specific mechanisms of action.

Response: Thanks for the point. We totally agree with the reviewer that further experiments are necessary to elucidate its specific mechanisms of action. Currently, we are actively engaged in conducting experiments to elucidate the underlying mechanisms by which *L*-aspartic acid induces intestinal fibrosis. However, the specific mechanism is out of the scope of the present study already with large amount of data. We anticipate presenting these findings in future reports; however, at

present, we can only acknowledge this as a limitation of our study (added in the Discussion section).

c) While the study investigated the promoting effect of *L*-aspartic acid on intestinal fibrosis, it did not mention the use of potential inhibitors or other interventions to reverse or slow down the fibrotic process. Testing known anti-fibrotic drugs or new compounds to see if they can counteract the effects of *L*-aspartic acid could be beneficial.

Response: We thank the reviewer's comment. This problem had been considered before conducting the experiment. In our experiment, TGF- β 1 was employed as an inducer for initiating intestinal fibrosis; however, it is worth noting that there are currently no efficacious pharmaceutical agents available for reversing intestinal fibrosis. Consequently, we were unable to employ any potential inhibitors or alternative interventions to reverse the fibrotic progress in our study. Despite this limitation, we proceeded with validation tests utilizing varying concentrations of *L*-aspartic acid and obtained results demonstrating its promotion of fibrosis (please refer to reviewer 3 comment 5a and Figure 6 in the main text).

Comment 6. Does the article discuss the limitations of the study, such as sample selection bias, potential confounding factors, and the generalizability of the results?

Response: We thank the reviewer's suggestion. We have added a discussion on the limitations of this study, including the aforementioned points raised by the reviewer (added in the Discussion section).

Comment 7. The overall structure of the article is clear, but certain sections may need further refinement to enhance readability.

a) In Figure 1d, the authors describe that Xylose exhibited significant abundance in BF2 patients, but without discussing the role of Xylose in intestinal fibrosis in the manuscript. Additionally, it is difficult to discern from Figure 1d alone that Xylose was significantly more abundant in BF2 patients compared to the other two groups. Similarly, Figure 1e does not make it clear that *L*-aspartic acid levels in blood samples specifically decreased significantly in the BF2 group. To improve clarity, the authors could have included p-values alongside the FDR in Figures 1c-e to better illustrate the statistical significance between the three groups.

Response: We thank the reviewer's comments. In Fig. 1d, Xylose exhibited significant abundance in BF2 patients in our study. However, to our best knowledge, its association with intestinal fibrosis has not been reported and requires further investigation into its role in the pathogenesis of intestinal fibrosis. We have added this information in the Results section of the main text (as shown in the fourth paragraph in the Results section).

Additionally, the selection of targets in our study was based on a variety of well-designed methodologies. These included not only comparative analysis between patients with BF1 and BF2, but also other investigations such as the correlation analyses between the multi-omics data and the macro-morphology detected on MR enterography (MRE). Although *L*-aspartic acid was not selected in the initial comparative analysis between BF1 and BF2 (hence it is not shown in Fig. 1e), its significance in intestinal fibrosis became evident after incorporating MRE which accurately depicts fibrosis-induced alterations in the intestine. In the subsequent integrated analysis of gut microbiota, metabolites, and MRE findings, *L*-aspartic acid demonstrated its important mediating effects in intestinal fibrosis, thereby confirming its role as a significant target for intestinal fibrosis.

The *P* value for the comparison of groups in Xylose in Fig. 1d is 0.038, indicating its statistical

significance. In response to the reviewer’s suggestion, we have included *P* values alongside the FDR in Figures 1c-e, as shown below. However, displaying both FDR and *P* values simultaneously on the figures may result in a cluttered appearance. Hence, we kindly request permission from the editor and reviewer to only present the FDR on Figures 1c-e. The exact *P* values can be found in Table EV9-10, as shown in the corresponding results section of the main text.

Figure 1

b) The stool and blood samples should be clearly marked in Figure 4d and 4e, respectively. The authors can also describe the results of Figure 4e in the manuscript and explain the relationship between the predicted microbial pathways and fecal aspartate levels.

Response: We apologize for the missing labels in figure 4d and 4e. We have now provided clear annotations for blood and fecal samples in these two figures. In the fig.4e, we also observed a similar negative association between microbial pathway and aspartic acid levels in fecal samples, although without statistical significance ($P=0.400$, $r=-0.12$). According to the reviewer's suggestion, we have

described the result of Fig.4e in the Results section. As the data suggested, we observe negative correlations between KO00250 and aspartic acid concentrations in both fecal and blood samples, indicating that the gut microbiota processes a metabolizing functionality of aspartic acid. However, we notice that the association between KO00250 and aspartic acid concentration in fecal samples is not statistically significant. The inconsistent relationships between blood and fecal metabolites have been repeatedly reported in previous studies, probably due to the difference of compound transformation, metabolic absorption and chemical format^{1,2}. Further study is warranted to investigate the specific microbial taxonomy and gene functions which are involved in the aspartic acid metabolism process. We also added this into our limitation.

1. Deng, K., Xu, J. J., Shen, L., Zhao, H., Gou, W., Xu, F., ... & Chen, Y. M. (2023). Comparison of fecal and blood metabolome reveals inconsistent associations of the gut microbiota with cardiometabolic diseases. *Nature Communications*, *14*(1), 571.
2. Chen, L., van den Munckhof, I. C., Schraa, K., Ter Horst, R., Koehorst, M., van Faassen, M., ... & Kuipers, F. (2020). Genetic and microbial associations to plasma and fecal bile acids in obesity relate to plasma lipids and liver fat content. *Cell reports*, *33*(1).

17th Jul 2024

Dear Prof. Li,

Thank you for the submission of your revised manuscript to EMBO Molecular Medicine. I am pleased to inform you that we will be able to accept your manuscript pending the following final amendments:

- 1) Address minor point raised by the referee #2. Please make sure that all statements in the manuscript are accurate.
- 2) Authors: We note that you currently have 4 first authors and together with you, a total of 5 co-corresponding authors. Is that correct? Do you confirm equal contribution of these authors, able to take full responsibility for the paper and its content? While there is no limit per se to the number of first and co-corresponding authors, 4 first and 5 co-corresponding authors is rather rare, and may not reflect as intended to the community.
- 3) In the main manuscript file, please do the following:
 - Please address all comments suggested by our data editors listed below:
 - o Figure legends:
 1. Please define the annotated p values *****/**/*** as well as provide the exact p-values for the same in the legend of figure EV 2c-d; as appropriate.
 2. Please note that the exact p values are not provided in the legends of figures 1a; 5b-d; 6a-c, e-f; EV 2a-b; EV 3.
 3. Please indicate the statistical test used for data analysis in the legends of figures 1b; 4a-b, d-e; EV 2a-b; EV 3.
 4. Please note that the box plot needs to be defined in terms of minima, maxima, centre, bounds of box and whiskers, and percentile in the legend of figure 1d.
 5. Please note that information related to n is missing in the legends of figures 1c-e; 2e-h.
 6. Although 'n' is provided, please describe the nature of entity for 'n' in the legend of figure 1a.
 - Add callouts for each panel in Fig EV1-3.
 - Author contributions: Please remove it from the manuscript and specify author contributions in our submission system. CRediT has replaced the traditional author contributions section because it offers a systematic machine-readable author contributions format that allows for more effective research assessment. You are encouraged to use the free text boxes beneath each contributing author's name to add specific details on the author's contribution. More information is available in our guide to authors:
<https://www.embopress.org/page/journal/17574684/authorguide#authorshipguidelines>
 - Please include structured Methods section that includes a Reagents and Tools Table followed by a Methods and Protocols section. More information on how to adhere to this format as well as downloadable templates (.docx) for the Reagents and Tools Table can be found in our author guidelines: <https://www.embopress.org/page/journal/17574684/authorguide#structuredmethods>
An example of a paper with Structured Methods can be found here:
<https://www.embopress.org/doi/full/10.1038/s44320-024-00037-6#sec-4>
 - In Methods, remove the sentence "All authors had access to the study data and had reviewed and approved the final manuscript."
 - In Methods, provide dilutions for each antibody used in the study.
 - In Methods, provide the statement that informed consent was obtained from all human subjects and confirm that the experiments conformed to the principles set out in the WMA Declaration of Helsinki and the Department of Health and Human Services Belmont Report.
 - Indicate in legends number and nature of replicates and exact p= values, not a range, along with the statistical test used. To keep the figures "clear" some authors found providing an Appendix table Sx with all exact p-values preferable. You are welcome to do this if you want to.
 - Remove DOIs from references.
 - In data availability statement please remove the sentence "All other data generated or analyzed during this study are included in this published article (and its supplementary information files)."
- 4) Expanded View: Please rename Tables EV2-7 and EV17-22 to Dataset EV1-12. Tables EV1 and EV8-16 should be renamed to Tables EV1-10. Update their callouts in the main manuscript text. Also, please rename EV Classifier to Computer Code EV1 and update its callout in the text.
- 5) Funding: Please make sure that information about all sources of funding are complete in both our submission system and in the manuscript. 2023B151502007 is currently missing in our submission system.
- 6) Synopsis:
 - Synopsis image: Please resize the image to 550 px-wide x (250-400)-px high and upload it as a high-resolution jpeg file.
 - Please check your synopsis text and image before submission with your revised manuscript. Please be aware that in the proof stage minor corrections only are allowed (e.g., typos).
- 7) For more information: Please place it before references.
- 8) As part of the EMBO Publications transparent editorial process initiative (see our Editorial at <http://embomolmed.embopress.org/content/2/9/329>), EMBO Molecular Medicine will publish online a Review Process File (RPF) to accompany accepted manuscripts. This file will be published in conjunction with your paper and will include the anonymous referee reports, your point-by-point response and all pertinent correspondence relating to the manuscript. Let us know whether you agree with the publication of the RPF and as here, if you want to remove or not any figures from it prior to publication.

9) Please provide a point-by-point letter INCLUDING my comments as well as the reviewer's reports and your detailed responses (as Word file).

I look forward to reading a new revised version of your manuscript as soon as possible.

Yours sincerely,

Zeljko Durdevic

*** Instructions to submit your revised manuscript ***

1) a .docx formatted version of the manuscript text (including Figure legends and tables)

2) Separate figure files*

3) supplemental information as Expanded View and/or Appendix. Please carefully check the authors guidelines for formatting Expanded view and Appendix figures and tables at <https://www.embopress.org/page/journal/17574684/authorguide#expandedview>

4) a letter INCLUDING the reviewer's reports and your detailed responses to their comments (as Word file).

5) The paper explained: EMBO Molecular Medicine articles are accompanied by a summary of the articles to emphasize the major findings in the paper and their medical implications for the non-specialist reader. Please provide a draft summary of your article highlighting

This may be edited to ensure that readers understand the significance and context of the research.

Please refer to any of our published articles for an example.

6) For more information: There is space at the end of each article to list relevant web links for further consultation by our readers. Could you identify some relevant ones and provide such information as well? Some examples are patient associations, relevant databases, OMIM/proteins/genes links, author's websites, etc...

7) Author contributions: the contribution of every author must be detailed in a separate section.

8) EMBO Molecular Medicine now requires a complete author checklist (<https://www.embopress.org/page/journal/17574684/authorguide>) to be submitted with all revised manuscripts. Please use the

checklist as guideline for the sort of information we need WITHIN the manuscript. The checklist should only be filled with page numbers where the information can be found. This is particularly important for animal reporting, antibody dilutions (missing) and exact values and n that should be indicated instead of a range.

9) Every published paper now includes a 'Synopsis' to further enhance discoverability. Synopses are displayed on the journal webpage and are freely accessible to all readers. They include a short stand first (maximum of 300 characters, including space) as well as 2-5 one sentence bullet points that summarise the paper. Please write the bullet points to summarise the key NEW findings. They should be designed to be complementary to the abstract - i.e. not repeat the same text. We encourage inclusion of key acronyms and quantitative information (maximum of 30 words / bullet point). Please use the passive voice. Please attach these in a separate file or send them by email, we will incorporate them accordingly.

You are also welcome to suggest a striking image or visual abstract to illustrate your article. If you do please provide a jpeg file 550 px-wide x 300-600px high.

10) A Conflict of Interest statement should be provided in the main text

11) Please note that we now mandate that all corresponding authors list an ORCID digital identifier. This takes <90 seconds to complete. We encourage all authors to supply an ORCID identifier, which will be linked to their name for unambiguous name identification.

Currently, our records indicate that the ORCID for your account is 0000-0002-7476-2644.

Link Not Available

12) Include a Reagents and Tools Table as part of the Methods section, which can be downloaded from our author guidelines (<https://www.embopress.org/page/journal/17574684/authorguide#structuredmethods>)

Photos 400-800 DPI

*Additional important information regarding figures and illustrations can be found at <https://bit.ly/EMBOPressFigurePreparationGuideline>. See also figure legend preparation guidelines: <https://www.embopress.org/page/journal/17574684/authorguide#figureformat>

***** Reviewer's comments *****

Referee #2 (Remarks for Author):

An excellent study which was improved further by the suggestions of the other reviewer.

Minor point:

From manuscript: For example, aspartate protease, an important biosynthetic product derived from L-aspartic acid, participates in various pathological processes, including the promotion of fibroblast proliferation. Reference:31.

My understanding is that aspartic proteases are catalytic protease enzymes rather than biosynthetic products derived from L-aspartate.

Referee #3 (Remarks for Author):

The authors have done an excellent job addressing my comments; I am satisfied with this revised manuscript and congratulate

the authors on an exceptional study.

Comments from the Reviewers:**Reviewer 2** (Remarks for Author):

An excellent study which was improved further by the suggestions of the other reviewer.

Minor point:

From manuscript: For example, aspartate protease, an important biosynthetic product derived from *L*-aspartic acid, participates in various pathological processes, including the promotion of fibroblast proliferation. Reference:31.

My understanding is that aspartic proteases are catalytic protease enzymes rather than biosynthetic products derived from *L*-aspartate.

Response: We would like to express our sincere gratitude for your recognition of our work and your insightful observation. Based on your feedback, we have thoroughly investigated the relevant literature and fully agree with your comment. To enhance the accuracy of our discussion of findings, we have deleted this sentence from the manuscript to eliminate any potential ambiguity.

Reviewer 3 (Remarks for Author):

The authors have done an excellent job addressing my comments; I am satisfied with this revised manuscript and congratulate the authors on an exceptional study.

Response: Thank you for your encouraging feedback on our revised manuscript. We greatly appreciate your recognition of our efforts, and we are grateful for your valuable insights, which have significantly improved our study.

13th Aug 2024

Dear Prof. Feng,

We are pleased to inform you that your manuscript is accepted for publication and is now being sent to our publisher to be included in the next available issue of EMBO Molecular Medicine.
